# Using Platt's scaling for calibration after undersampling – limitations and how to address them

**Nathan Phelps**  *nphelps3@uwo.ca*
*Department of Statistical and Actuarial Sciences*
*University of Western Ontario*

**Daniel J. Lizotte**  *dlizotte@uwo.ca*
*Department of Computer Science*
*Department of Epidemiology and Biostatistics*
*University of Western Ontario*

**Douglas G. Woolford**  *dwoolfor@uwo.ca*
*Department of Statistical and Actuarial Sciences*
*University of Western Ontario*

**Reviewed on OpenReview:** *https://openreview.net/forum?id=80b2zaeTUe*

## Abstract

When modelling data where the response is dichotomous and highly imbalanced, response-based sampling where a subset of the majority class is retained (i.e., undersampling) is often used to create more balanced training datasets prior to modelling. However, the models fit to this undersampled data, which we refer to as base models, generate predictions that are severely biased. There are several calibration methods that can be used to combat this bias, one of which is Platt's scaling. Here, a logistic regression model is used to model the relationship between the base model's original predictions and the response. Despite its popularity for calibrating models after undersampling, Platt's scaling was not designed for this purpose. Our work presents what we believe is the first detailed study focused on the validity of using Platt's scaling to calibrate models after undersampling. We show analytically, as well as via a simulation study, that Platt's scaling should not be used for calibration after undersampling without critical thought. If Platt's scaling would have been able to successfully calibrate the base model had it been trained on the entire dataset (i.e., without undersampling), then Platt's scaling might be appropriate for calibration after undersampling. If this is not the case, we recommend a modified version of Platt's scaling that fits a logistic generalized additive model to the logit of the base model's predictions, as this method is theoretically motivated and performed relatively well across the settings considered in our study.

## 1 Introduction

Highly imbalanced, binary classification problems are ubiquitous in today's world of data analytics. These problems appear in fields as varied as finance (e.g., fraud detection; Varmedja et al. 2019), health care (e.g., disease modelling; Shin et al. 2023), and wildfire (e.g., fire occurrence prediction; Phelps & Woolford 2021a). In our study, we will assume that the minority class represents the occurrence of the outcome of interest (denoted by 1; also called the positive class), such as a fraudulent transaction, and the majority class represents its non-occurrence (denoted by 0; also called the negative class). Oftentimes, the datasets associated with imbalanced classification problems are very large (i.e., millions—or more—of observations with many covariates). Both the imbalance and size of these datasets can present modelling challenges. Fitting complex models (e.g., deep learning models) to massive datasets can be very time-consuming and, in

addition, studies have shown that some models tend to neglect the minority class in the face of substantial class imbalance (e.g., Japkowicz & Stephen 2002). A common method for handling both these issues is undersampling, or downsampling (e.g., Wallace & Dahabreh 2014; Moreau et al. 2020; Peng et al. 2020; Phelps & Woolford 2021a; Burmeister et al. 2023; Shin et al. 2023).

When undersampling, all observations from the minority class are kept but only a random subset of the majority class is retained for modelling. The problem with this sampling procedure is that it biases the model. Consider a data distribution $f_{(\mathbf{X},Y)}(\mathbf{x}, y)$, where $\mathbf{x}$ is a vector of covariates and $y$ is a binary outcome, and a model $h$ that produces estimates $h(\mathbf{x}) \approx \mathbb{P}(Y = 1|\mathbf{X} = \mathbf{x})$. We define a perfectly calibrated model as one that generates estimates such that for any $\hat{p} = h(\mathbf{x})$ that can be produced by the model, $\mathbb{P}(Y = 1|h(\mathbf{X}) = \hat{p}) = \hat{p}$, where the probability is computed over the data distribution $f_{(\mathbf{X},Y)}$. For example, for observations where the model assigns $\hat{p} = 0.3$, we expect that 30% are truly positives. Because the distribution of the training data now differs from that of new data, a model $h$ that was well-calibrated on its undersampled dataset will not be well-calibrated when used to make predictions on new data. Generally, a model trained on undersampled data will output estimates that overestimate the true outcome probabilities. This bias induced by undersampling is a serious issue because having poorly calibrated probability estimates can hinder the effectiveness of the model for use in practice, altering prevalence estimates and potentially leading to suboptimal decision-making (e.g., Phelps & Woolford 2021b; Guilbert et al. 2024). Consequently, it is important to adjust the probability estimates of the model to try to obtain well-calibrated estimates whenever possible.

There are several different methods for calibrating models after undersampling. One of the most common approaches is Platt's scaling (Platt 1999). Despite its popularity in this situation (e.g., Wallace & Dahabreh 2014; Moreau et al. 2020; Peng et al. 2020; Phelps & Woolford 2021a; Burmeister et al. 2023; Shin et al. 2023), Platt's scaling was originally designed for another purpose: augmenting the output of support vector machines to obtain calibrated probabilities. It has since been used to calibrate models in other situations, sometimes because models were trained on undersampled data (e.g., Wallace & Dahabreh 2014; Moreau et al. 2020; Peng et al. 2020; Phelps & Woolford 2021a; Burmeister et al. 2023; Shin et al. 2023) and sometimes because models were miscalibrated for other reasons (e.g., Guo et al. 2017; Ojeda et al. 2023). Platt's scaling involves fitting a logistic regression model, enforcing a sigmoid relationship between covariates and probabilities. Because of this restriction, the validity of using Platt's scaling outside of its original purpose has been debated; some have criticized it (e.g., Naeini et al. 2015; Kull et al. 2017), but Böken (2021) showed that it is justifiable to use Platt's scaling in more scenarios than other works have suggested. From what we have seen in the literature involving undersampling, it does not appear that the appropriateness of using Platt's scaling in the context of calibration after undersampling is well-understood.

Our work presents what we believe is the first detailed study on the validity of using Platt's scaling to calibrate models after undersampling. First, we analytically show that Platt's scaling is incapable of properly calibrating a model perfectly fit to an undersampled dataset, leading to incorrect estimates of conditional probabilities. We also show how Platt's scaling can be modified so that it can provide correct estimates. Next, we demonstrate that traditional Platt's scaling can be effective when the model fit to undersampled data has a specific systematic error. Finally, we consider another adjustment for improving performance in more general settings.

In Section 2, we outline the calibration approaches considered in this study and provide our primary theoretical results; Section 3 considers wildland fire occurrence prediction as motivation for our work; Section 4 details the simulation study we conducted; and, in Section 5, we provide conclusions and practical recommendations.

## 2   Calibration after undersampling

We consider calibrating binary classification models fit to undersampled data. In this context, we have a dichotomous response random variable $Y$ that takes values 0 (i.e., the negative class) or 1 (i.e., the positive class) and associated covariates $\mathbf{X}$, and our goal is to estimate $p(\mathbf{x}) = \mathbb{P}(Y = 1|\mathbf{X} = \mathbf{x}) \in (0, 1)$. We also introduce a dichotomous variable $S$ that indicates which observations will be used for estimating $p(\mathbf{x})$ by first estimating $\gamma(\mathbf{x}) = \mathbb{P}(Y = 1|\mathbf{X} = \mathbf{x}, S = 1) \in (0, 1)$ and then transforming $\gamma$ using a calibration function $\kappa$, giving $p(\mathbf{x}) = \kappa(\gamma(\mathbf{x}))$. Note that if $S$ is independent of $\mathbf{X}$ and $Y$ (e.g., if we take a simple random sample

of the training set), then $\kappa$ is the identity function. However, when undersampling, we keep all positive instances (i.e., $S = 1$ for all positive instances) and negative instances are kept only with probability $\pi_0$. We refer to the model that learns from the undersampled training dataset as the base model, and its predictions as $\hat{\gamma}$'s. If the calibration function $\kappa$ is learned using another model, we refer to this model as the calibration or secondary model. This approach is an instance of model stacking, whereby a meta learner (the calibration model) is trained based on outputs of one or more base models.

## 2.1 Platt's scaling and its variations

Platt's scaling (Platt 1999) calibrates the predictions of the base model by fitting a logistic regression model to the $y$'s using the $\hat{\gamma}$'s as the predictor. In the original paper, Platt (1999) slightly modified the responses to perform regularization (see details in Appendix 1), but Platt's scaling has often been implemented without regularization (e.g., Phelps & Woolford 2021a; Ojeda et al. 2023). It is therefore sometimes called logistic calibration (e.g., Kull et al. 2017), although Ojeda et al. (2023) used this terminology to refer to a different model. In our work, we leave the responses as 0/1 variables. When using logistic regression, we assume that $Y \sim \text{Bernoulli}(p)$ and that there is a linear relationship between the logit of the $p$'s and the $\hat{\gamma}$'s (see Eq. 1).

$$\log\left(\frac{p}{1-p}\right) = \beta_0 + \beta_1\hat{\gamma} \tag{1}$$

After fitting the logistic regression model (i.e., learning estimates $\hat{\beta}_0$ and $\hat{\beta}_1$ for $\beta_0$ and $\beta_1$, respectively), we obtain the following calibration model from this approach:

$$\kappa(\hat{\gamma}) = \frac{\exp(\hat{\beta}_0 + \hat{\beta}_1\hat{\gamma})}{1 + \exp(\hat{\beta}_0 + \hat{\beta}_1\hat{\gamma})} \tag{2}$$

To assess the validity of Platt's scaling as a method for calibration after undersampling, we must determine if the relationship assumed by this approach is reasonable. Naturally, this depends on the base model, since it outputs the $\hat{\gamma}$'s. We first consider the case where the base model is perfect, resulting in Theorem 1.

**Theorem 1.** *If the base model provides perfect estimates based on the undersampled training dataset, Platt's scaling is incorrectly specified and cannot provide perfect probability estimates on the full dataset.*

**Proof:** First, we need to determine the relationship between a base model's predictions ($\hat{\gamma}$'s) and the corresponding true probabilities ($p$'s). Since we are considering a perfect base model, $\hat{\gamma} = \gamma$. As a result, we can refer to previous literature (Dal Pozzolo et al. 2015b), which has shown the following:

$$p = \frac{\gamma\pi_0}{1 - \gamma + \gamma\pi_0} \tag{3}$$

We can then substitute Eq. 3 for $p$ into Eq. 1 to determine the true relationship between $p$ and $\gamma$ on the log odds scale. After some algebra, the result of this substitution is Eq. 4.

$$\log\left(\frac{p}{1-p}\right) = \log(\pi_0) + \log\left(\frac{\gamma}{1-\gamma}\right) \tag{4}$$

Clearly, the relationship between the $p$'s and the $\gamma$'s is not linear on the log odds scale. Thus, even though the base model perfectly modeled its training dataset, the secondary model that will be learned from Platt's scaling cannot possibly properly adjust the $\hat{\gamma}$'s to achieve a perfect final model. $\square$

For those familiar with both the literature on Platt's scaling and on undersampling, the result of Theorem 1 might be expected. One of the criticisms of Platt's scaling is its inability to leave a perfect model perfectly calibrated (e.g., Kull et al. 2017), and logistic models trained on undersampled data are calibrated through only adjusting their intercept (e.g., Taylor et al. 2013; Phelps & Woolford 2021a), so it is intuitive that Platt's scaling is unable to calibrate models that are perfect with respect to the undersampled data. However,

Platt's scaling's inability to calibrate such models is more disguised in this setting, as Platt's scaling generally will still improve calibration due to the base model's extreme overprediction because of learning from the undersampled dataset (e.g., Phelps & Woolford 2021b).

Theorem 1 shows that Platt's scaling cannot properly calibrate a perfect base model, but a corollary of Eq. 4 is that a simple transformation can be done to remedy this problem. Rather than using $\hat{\gamma}$ as the covariate, we can use $\log\{\hat{\gamma}/(1-\hat{\gamma})\}$. This transformation has been considered in the calibration literature before, such as by Turner et al. (2014) for linear in log odds calibration, by Leathart et al. (2017) for fitting probability calibration trees (which involve fitting a logistic regression model in the leaf nodes), and by Kull et al. (2017), who call this beta$[a = b]$ calibration (see Section 2.2.2 for more details). Böken (2021) noted that Platt's scaling was designed for a predictor that can take any real value, so it makes sense to use this transformation to convert predictions from $[0, 1]$ to the real line. Via their simulation study (which did not incorporate undersampling), Ojeda et al. (2023) found that using the logit transformation generally improved calibration. Although this approach has appeared several times in the calibration literature, we are not aware of any work that has explicitly theoretically motivated its use for calibration after undersampling. Of course, we cannot expect the base model to be perfect in practice, but we expect that defining the logistic regression model using this transformation of $\hat{\gamma}$ will still be effective for models where $\hat{\gamma} \approx \gamma$.

Models sometimes exhibit systematic estimation errors (e.g., Niculescu-Mizil & Caruana 2005; Guo et al. 2017; Guilbert et al. 2024). Some models, such as random forests and boosted decision trees, have been shown to push probability estimates towards 0.5, while models like Naïve Bayes push estimates towards extreme values (i.e., 0 or 1). We now consider models where $\hat{\gamma}$ and $\gamma$ have a sigmoidal relationship, represented by Eq. 5.

$$\gamma = \frac{1}{1 + \exp[-r(\hat{\gamma} - m)]} \tag{5}$$

Here, $r$ and $m$ are arbitrary constants. It is worth noting, however, that there are values of $r$ and $m$ that do not lead to a reasonable representation of a model (e.g., $r < 0$ and $m \in [0, 1]$ leads to a model whose predicted probabilities are negatively related to the true probabilities). Under settings that do reasonably represent a model (e.g., $r = 10$ and $m = 0.5$), this relationship represents a model whose estimates are pushed towards 0.5. Calibrating models that err in the form indicated by Eq. 5 is considered a valid use of Platt's scaling (e.g., Kull et al. 2017; Leathart et al. 2017) because the logistic regression model's parametric assumptions are met. In Theorem 2, we show that these assumptions are still met when the base model learns from an undersampled dataset.

**Theorem 2.** *If the base model provides predictions that have a sigmoidal relationship with the true probabilities on the undersampled training dataset, Platt's scaling is correctly specified and can provide perfect probability estimates on the full dataset.*

**Proof:** We can substitute Eq. 5 for $\gamma$ in Eq. 4. After some algebra, we can obtain Eq. 6.

$$\log\left(\frac{p}{1-p}\right) = \log(\pi_0) - rm + r\hat{\gamma} \tag{6}$$

The relationship between the logit of the $p$'s and the $\hat{\gamma}$'s is now linear, so the assumption of the logistic regression model is met. A logistic regression model with the learned coefficients $\hat{\beta}_0 = \log(\pi_0) - rm$ and $\hat{\beta}_1 = r$ would perfectly calibrate this base model. $\square$

We have addressed how models can be calibrated after undersampling if the model is perfect or if its predictions have a sigmoid shape with the true probabilities, but we have not addressed models with a tendency to push estimates towards extreme values or models that deviate somewhat from either perfect prediction or a perfect sigmoid shape. In both cases, we cannot derive a simple transformation that will allow us to satisfy the assumptions of logistic regression. However, this does not preclude us from being able to modify Platt's scaling to obtain better probability estimates. Rather than instituting the restrictive assumptions of logistic regression, a logistic Generalized Additive Model (GAM) can be used. This has

also been used for calibration before (e.g., Lucena 2018), including to calibrate models after undersampling in a few studies (e.g., Coussement & Buckinx 2011; Phelps & Woolford 2021a), but it does not seem to be a common approach for this purpose. Logistic GAMs use smoothers to model the relationship between covariates and the outcome, relaxing the assumption of linearity on the log odds scale so that non-linear relationships can be modeled. It is important to note that, given enough data, a logistic GAM will converge to a logistic regression model when the linearity assumption holds.

## 2.2 Alternative calibration methods

### 2.2.1 Analytical calibration

As we have seen in Eq. 3, Dal Pozzolo et al. (2015b) analytically computed the relationship between the $p$'s and the $\gamma$'s. Thus, a simple way to adjust for the bias induced by undersampling is to use this equation as a calibration function, with the predictions ($\hat{\gamma}$'s) in place of $\gamma$ as shown in Eq. 7.

$$\kappa(\hat{\gamma}) = \frac{\hat{\gamma}\pi_0}{1 - \hat{\gamma} + \hat{\gamma}\pi_0} \tag{7}$$

A benefit of analytical calibration is that it does not require learning the calibration function from another full (i.e., not undersampled) dataset, unlike the model-based approaches. In addition, if the base model is perfect (i.e., $\hat{\gamma} = \gamma$), then this analytical approach perfectly calibrates the probability estimates. However, the additional training time of model-based approaches is not typically a problem because it is not computationally expensive to learn a function based on only one covariate. This approach also does not account for any error in the base model, so it may struggle in practice when models are imperfect.

### 2.2.2 Beta calibration

As mentioned previously, the primary criticism of Platt's scaling is its restrictive parametric assumptions. Consequently, Kull et al. (2017) developed beta calibration. Beta calibration is still a parametric method, but it is more flexible than Platt's scaling. The calibration function learned in beta calibration is shown in Eq. 8. We have adopted the parameter names (i.e., $a$, $b$, and $c$) of Kull et al. (2017) but have expressed the calibration function differently to make it more comparable to Platt's scaling as presented herein.

$$\kappa(\hat{\gamma}) = \frac{\exp[c + a\log(\hat{\gamma}) - b\log(1 - \hat{\gamma})]}{1 + \exp[c + a\log(\hat{\gamma}) - b\log(1 - \hat{\gamma})]} \tag{8}$$

A special case of beta calibration is beta[$a = b$] calibration. As mentioned in Section 2.1, this is equivalent to Platt's scaling with the logit transformation with $\hat{\beta}_0 = c$ and $\hat{\beta}_1 = a = b$. Throughout our study, we will refer to Platt's scaling with the logit transformation as a variation of Platt's scaling, but it should be noted that it could also be referred to as a variation of beta calibration.

Kull et al. (2017) showed that beta calibration performed similarly to Platt's scaling when Platt's scaling's assumptions were approximately met (i.e., when calibrating models that pushed probability estimates towards 0.5). When the assumptions of Platt's scaling were not met, beta calibration clearly outperformed Platt's scaling. However, that study did not focus on calibrating models after undersampling.

### 2.2.3 Isotonic regression

Isotonic regression (e.g., Zadrozny & Elkan 2002) is the most flexible calibration method we consider in this study. It minimizes $\sum_{i=0}^{n} [y_i - \kappa(\hat{\gamma}_i)]^2$, where $n$ is the size of the dataset and $\kappa$ is a step function. This model is learned using the pair-adjacent violators algorithm (Ayer et al. 1955), and the only restriction imposed on $\kappa$ is that it must be monotonically non-decreasing. This is a very flexible method, but because $\kappa$ is a step function it does not produce a smooth relationship between the base model's predictions and the new probability estimates.

# 3    A motivating example: Wildland fire occurrence prediction

Wildland fire occurrence prediction is an important part of a fire management agency's planning. Often, the study region and period is partitioned into a set of voxels whose resolution is fine enough that counts of fire occurrences can be modeled as a presence/absence problem. This results in an extremely imbalanced binary classification problem. For example, de Haan-Ward et al. (2024) had less than 0.1% positive cases in their study region of the province of Ontario, Canada. We also consider data from Ontario, provided to us by the Ontario Ministry of Natural Resources and spanning the years 2000 to 2019. Using 20km × 20km × daily voxels, the dataset provides information on the day of year, location, and the weather, including both standard weather variables (e.g., temperature) and fire-weather variables (e.g., Fine Fuel Moisture Code), as well as variables describing land use. To avoid the effects of a potential temporal trend, we split the data as follows: years 2000, 2003, 2006, 2009, 2010, 2013, 2016, and 2019 were used for training; years 2001, 2004, 2007, 2011, 2014, and 2017 were used for calibration; and years 2002, 2005, 2008, 2012, 2015, and 2018 were used for testing. All three datasets had over 2.2 million observations. We would not recommend our splitting procedure for operational use, where detecting temporal trends would be important.

For our example, our base model is a logistic GAM that predicts the occurrence of human-caused fires using smoothers for day of year, temperature, relative humidity, wind speed, rain, Fine Fuel Moisture Code, and area of each of infrastructure interface, urban interface, and industrial interface, as well as longitude and latitude (as a bivariate smoother). To create the training dataset, we kept all fire observations and 0.2% of the non-fire observations, leading to just over a quarter of the observations being fires. A GAM is useful for illustrative purposes in this setting because we can compare the predictions obtained from various calibration approaches to the predictions obtained from the well-established method of adjusting the intercept of the GAM (e.g., Taylor et al. 2013; Phelps & Woolford 2021a). Adding an offset of $\log(\pi_0)$ to the intercept learned from the undersampled dataset has been shown to account for the bias induced by the sampling procedure. Note that this offset is equivalent to using analytical calibration for the GAM.

The calibration methods we considered here are Platt's scaling and Platt's scaling with the logit transformation. Fig. 1 shows reliability plots for both approaches. These were constructed by creating bins of width 0.001 based on the predictions, then computing the average prediction and average rate of fire occurrence within each bin and plotting them. We plotted only bins with more than 50 observations to avoid extremely noisy results. We also added 95% prediction intervals, computed using the 2.5th and 97.5th quantiles obtained from simulating outcomes under the assumption that the predicted probabilities were correct. The results seem to show that Platt's scaling underestimates the probabilities for relatively large probabilities. When using the logit transformation, this systematic underestimation appears to be mitigated and potentially eliminated, although the noisiness in the data makes it difficult to make definitive conclusions.

**Panel A**                                      **Panel B**

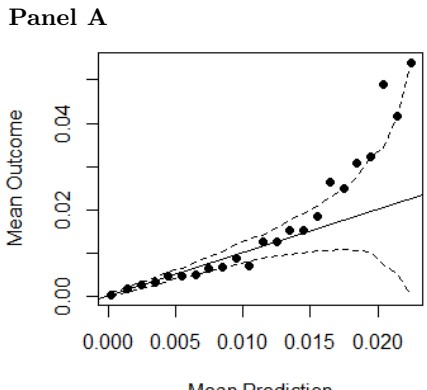
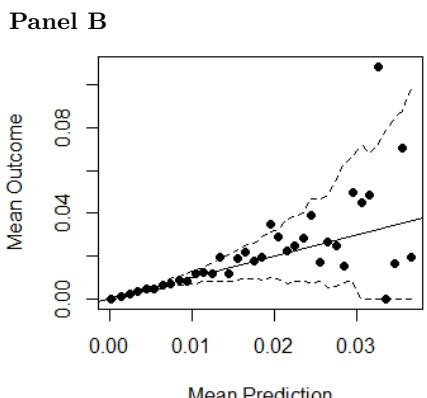

Figure 1: Reliability plots for a logistic Generalized Additive Model (GAM) calibrated using Platt's scaling (Panel A) and Platt's scaling with the logit transformation (Panel B). The solid line is the 45° line and the dashed lines represent 95% prediction intervals, computed assuming the predicted probabilities from the modelling were correct. Bins with 50 or less observations were removed.

It should be noted that plotting only bins with more than 50 observations led to removing the 48 observations with an estimate greater than 0.023 for traditional Platt's scaling (none of which were truly fires) and removing the 362 observations with an estimate greater than 0.037 for Platt's scaling with the logit transformation. The mean prediction for these observations was 0.047, while the true mean occurrence was 0.044. Versions of the plots in Fig. 1 but with all the data are available in Appendix 2.

Fig. 2 has line plots comparing the predictions of the two Platt's scaling approaches to the predictions from the model with an offset. The plots show that Platt's scaling and the offset lead to very different predictions; the offset generates much larger predictions. This is consistent with the underprediction observed in the reliability plot. On the other hand, the predictions from Platt's scaling with the logit transformation and the offset are nearly identical, suggesting that the logit transformation has worked well.

**Panel A**     **Panel B**

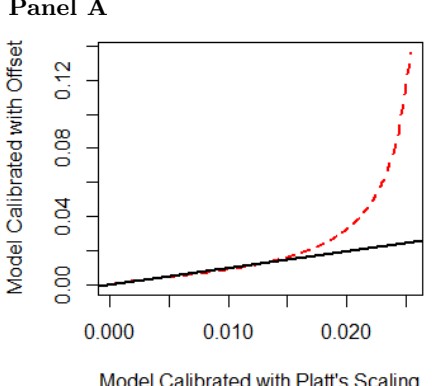
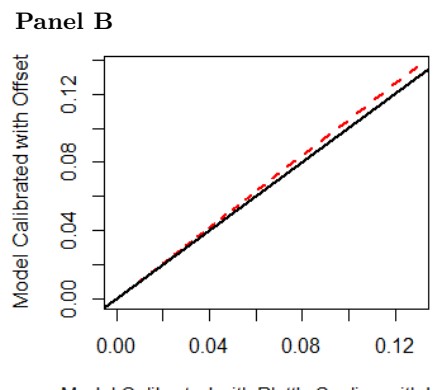

Figure 2: Plots comparing the predictions of the model with an offset to the model without an offset calibrated using Platt's scaling (Panel A) and Platt's scaling with the logit transformation (Panel B). The solid line is the 45° line and the dashed line shows the relationship between the predictions.

Although it appears that Platt's scaling leads to underestimates of the probability of a fire for the higher probability cases and that this may be fixed by using the logit transformation, it is difficult to make definitive conclusions. The rightmost points on the reliability plots are subject to high variability, so our conclusion that the logit transformation has helped is based largely on how well the predictions from that approach align with the predictions from using the offset. Thus, a simulation study can provide useful insights here. We can simulate as much data as needed and, more importantly, the true probabilities would be known, so we need not rely on reliability plots or comparisons with other methods—we can directly compare the estimates from each method to the true probabilities.

## 4 Simulation study

### 4.1 Methods

In this section, we describe the simulation study we used to evaluate the four variations of Platt's scaling outlined in Section 2.1: traditional Platt's scaling, Platt's scaling with the logit transformation, Platt's scaling with a logistic GAM, and Platt's scaling with both the logit transformation and a logistic GAM. These methods were compared to analytical calibration, beta calibration, and isotonic regression. Our simulation study was implemented in R (R Core Team 2023).

#### 4.1.1 Simulating data

To compare the various calibration methods, we simulated datasets with known outcome probabilities. We defined a success as belonging to the positive class and considered three different data generating processes, each with different mean probabilities of success. Through tweaking a parameter in our data generating process (see Appendix 3 for details), we obtained data generating processes with mean success probabilities of

approximately 0.0022, 0.0208, and 0.1109. These differing settings allowed us to study the effects of different levels of class imbalance. Although the levels of class imbalance differ, all are amenable to undersampling.

We were also interested in the effects of the size of the calibration dataset, so we considered large datasets differing in size by an order of magnitude. Our smaller calibration dataset had 100 000 observations and our larger dataset had 1 000 000 observations. To try to mitigate the effects of random variability in our results, our testing datasets all had 1 000 000 observations.

### 4.1.2 Simulating model outputs

Rather than modelling the output of our simulated datasets as a function of the covariates, we defined hypothetical base models with various estimation errors. The first hypothetical base model we considered is a perfect model (i.e., $\hat{\gamma} = \gamma$). Note that this model still required calibration to account for the undersampling process. Next, we considered models whose generated $\hat{\gamma}$'s systematically deviate from the $\gamma$'s. As discussed previously, systematic error has been found in very well-known models; random forests and boosted trees have been shown to push probability estimates towards 0.5, while Naïve Bayes models have been shown to push estimates towards extreme values (i.e., 0 or 1) (e.g., Niculescu-Mizil & Caruana 2005; Guilbert et al. 2024). We have represented models that push probability estimates towards 0.5 with the relationship shown in Eq. 9. Models that push probability estimates towards extreme values are represented by the relationship in Eq. 10. Plots illustrating each of these relationships are available in Appendix 3.

$$\hat{\gamma} = \min\left[\max\left(-\frac{1}{10}\log\left(\frac{1}{\gamma}-1\right)+0.5, 0\right), 1\right] \tag{9}$$

$$\hat{\gamma} = \frac{1}{1+\exp[-10(\gamma-0.5)]} \tag{10}$$

Finally, we considered a model that generates nearly perfect predictions in expectation, but does make errors. This was implemented by incorporating noise on the log odds scale of the base model. This noise was added via a normally distributed random variable with a mean of zero and standard deviation of 0.2. Because the datasets are imbalanced, this results in $\hat{\gamma}$'s with a mean slightly larger than the mean of the $\gamma$'s. A scatterplot showing the relationship between the $\hat{\gamma}$'s and the $\gamma$'s when the mean outcome probability is 0.0208 is shown in Appendix 3.

To obtain the $\gamma$'s for each data generating process, we set sampling rates that would generate approximately balanced training datasets. For the data generating processes described in Section 4.1.1, we used sampling rates of $\pi_0 = 0.0023$, $\pi_0 = 0.02125$, and $\pi_0 = 0.125$.

### 4.1.3 Implementing and evaluating the calibration methods

Analytical calibration was implemented using only base R. To implement the four variations of Platt's scaling, we used the glm and gam functions from the stats and mgcv packages (Wood 2011), respectively. Beta calibration was implemented using the betacal package (Kull et al. 2021), with three free parameters to be learned. We found that this sometimes resulted in an error, so we implemented beta$[a = b]$ calibration (i.e., Platt's scaling with the logit transformation) whenever this occurred. Like Platt's scaling, isotonic regression was implemented using functions from the stats package. For each isotonic regression model, we fit the model then converted it to a step function to make predictions on the testing dataset.

In our study, we paid special attention to the ability of each calibration method to fit the true relationship between the $\hat{\gamma}$'s and the $p$'s. Unlike when working with real data, the $p$'s are known in our study. We can take advantage of this by creating line plots of the $\hat{p}$'s against the $p$'s to visually evaluate the calibration of the predictions across all probabilities. For the model whose predictions are nearly perfect in expectation, line plots do not lead to a clear curve because of the randomness in the model's errors. Instead, we created reliability plots, but used the true probabilities instead of the outcomes (as is done with real data) to eliminate the effects of noise in the outcome. Since we used the probabilities, we plotted bins with more than 10 observations (as opposed to more than 50 in Section 3). To avoid relying entirely on visual assessments,

we also measured the gap between the $\hat{p}$'s and the $p$'s using root mean squared error (RMSE) and mean absolute error (MAE). To account for variability in our results, we ran our simulations 10 additional times and computed the means across these runs, as well as 95% confidence intervals for the mean. We used different training and testing datasets for each run, so this process accounted for multiple sources of variability.

## 4.2 Results

### 4.2.1 Perfect base model

The line plots for each of our simulation settings revealed similar results, so we only show two representative plots (Fig. 3). All six figures are available in Appendix 4. Here, we omit analytical calibration because it is known that it will perfectly calibrate the predictions.

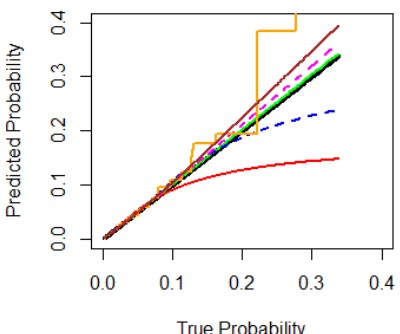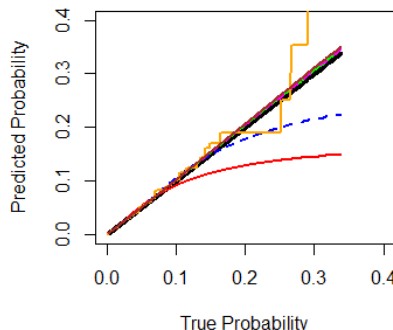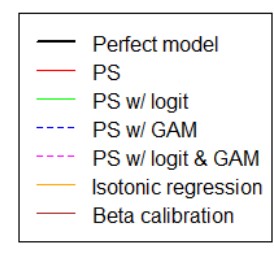

Figure 3: Probability estimates on the testing dataset from Platt's scaling (PS) and its variations, as well as isotonic regression and beta calibration. The data generating process has a mean probability of success of 0.0208. The base model is perfect and the calibration methods were trained on 100 000 observations (left) and 1 000 000 observations (right).

In Fig. 3, the lack of smoothness in the isotonic regression is very apparent. Although the model is flexible, its performance was quite poor, especially for the rarer cases with relatively high probabilities. Likewise, Platt's scaling struggled with these cases, systematically producing estimates that were far too low. These results confirm our analysis in Section 2.1, showing that Platt's scaling is unable to properly calibrate a perfect base model after undersampling. They also provide additional information about the way in which Platt's scaling fails in this setting: severe underestimation of success for higher probability outcomes, as was suspected from the results obtained on the wildland fire data in Section 3. Using the logit transformation, however, fixed this problem. With sufficient data, both Platt's scaling approaches with this transformation yielded nearly perfect predictions. Beta calibration also performed similarly to these two models. Of these three, Platt's scaling with the logit transformation generally had the best performance, but the differences were marginal (see Appendix 4). This is presumably due to the limited flexibility of the model, which is advantageous when the model is correctly specified (as it is here).

### 4.2.2 Base model pushes probability estimates towards 0.5

When the base model pushes probability estimates towards 0.5, the relative effectiveness of the calibration methods changed considerably (see Fig. 4 and additional results in Appendix 5). Notably, analytical calibration performed incredibly poorly because it does not account for the miscalibration of the base model at all. However, traditional Platt's scaling does account for this miscalibration, as shown in Section 2.1, resulting in it generally being the most effective approach with this base model. Platt's scaling with a GAM often resulted in very similar performance. Once again, beta calibration performed similarly to the two Platt's scaling approaches with the logit transformation, with all of them generally overestimating the probability of success for higher probability outcomes (see Fig. 4). However, of these three models, Platt's scaling with the logit transformation and a GAM was the clear best choice when the mean success probability

was 0.1109 and the calibration dataset had 1 000 000 observations. Here, the flexibility of the GAM allows it to better fit the relationship between the predictions and true probabilities.

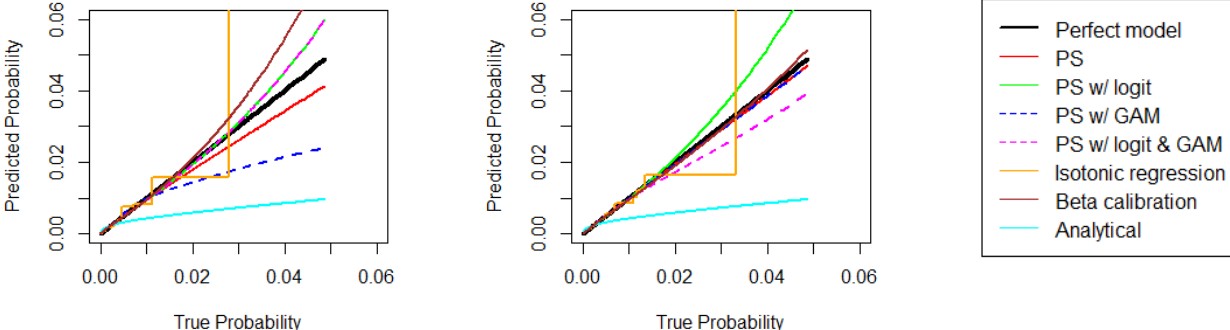

Figure 4: Probability estimates on the testing dataset from Platt's scaling (PS) and its variations, as well as isotonic regression, beta calibration, and analytical calibration. The data generating process has a mean probability of success of 0.0022. The base model pushes probability estimates towards 0.5 and the calibration methods were trained on 100 000 observations (left) and 1 000 000 observations (right).

### 4.2.3 Base model pushes probability estimates towards 0 or 1

When the base model pushes probability estimates towards extreme values, none of the calibration methods generated predictions that closely aligned with the true probabilities (see Fig. 5 and additional results in Appendix 6). Except for analytical calibration and isotonic regression, all the methods underestimated success probabilities for higher probability observations. Analytical calibration severely overestimated these probabilities and provided terrible estimates. Isotonic regression is the only calibration method that did not exhibit visible systematic biases with this base model. This is because isotonic regression's only assumption about the relationship between the predictions and true probabilities is that it is monotonically non-decreasing. However, its use of a step function still limits how well it can approximate the true probabilities. In terms of RMSE and MAE (see Appendix 6), Platt's scaling using the logit transformation and a GAM generally performed the best. Although Platt's scaling using the logit transformation and a GAM did systematically underestimate success probabilities for higher probability events, its underestimation was usually less severe than the other Platt's scaling approaches and beta calibration. For larger success rates and calibration datasets, isotonic regression was the second-best performing calibration method.

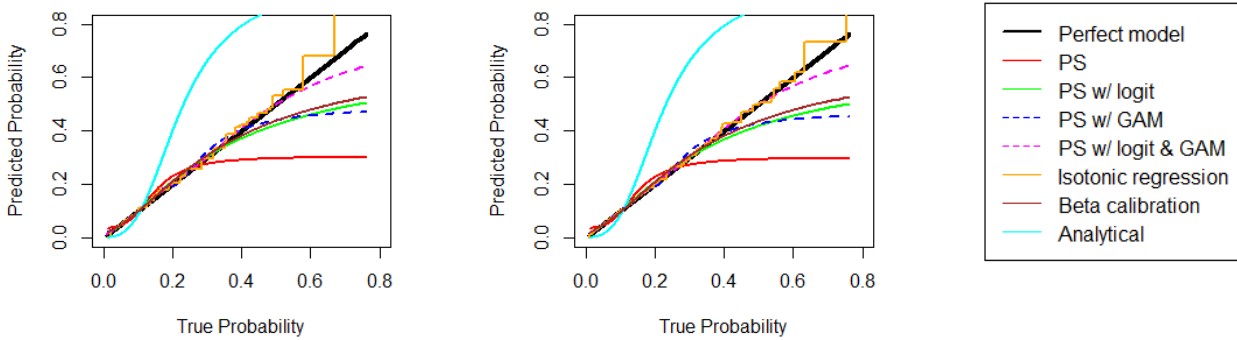

Figure 5: Probability estimates on the testing dataset from Platt's scaling (PS) and its variations, as well as isotonic regression, beta calibration, and analytical calibration. The data generating process has a mean probability of success of 0.1109. The base model pushes probability estimates towards 0 or 1 and the calibration methods were trained on 100 000 observations (left) and 1 000 000 observations (right).

#### 4.2.4 Base model is nearly perfect in expectation

For the base model that is nearly perfect in expectation, the line plots we've used thus far do not work because of the noise in the model's errors. Consequently, we created reliability plots instead. Note that these plots must be interpreted differently from the plots in Figs. 3 - 5. If the line is directly along the 45° line, it no longer means the calibration method generates correct probability estimates. Instead, it means that the model is perfectly calibrated. Recall that this means that $\mathbb{P}(Y = 1|h(\mathbf{X}) = \hat{p}) = \hat{p}$ for all $\hat{p}$, where $h$ is the model. The axes for reliability plots also differ from the axes in the earlier figures. With sufficient information, it seems that nearly all of the calibration methods lead to fairly well-calibrated predictions. The exception to this is Platt's scaling, which consistently produced poorly calibrated predictions. Using a GAM helped except for when the mean outcome success rate was 0.0022 (see Appendix 7). It should be noted that isotonic regression had some sets of larger predictions that are not shown in the plots.

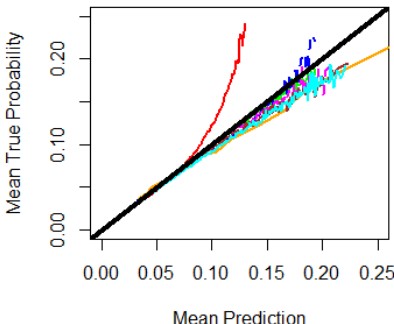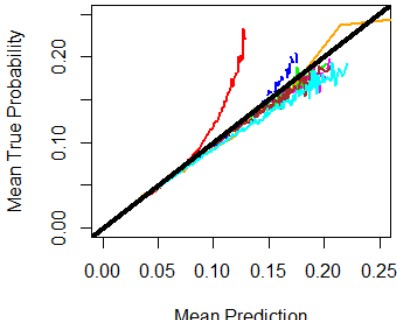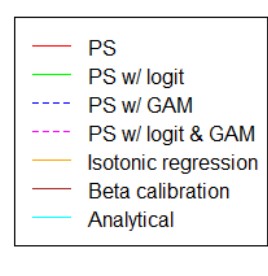

Figure 6: Reliability plots for Platt's scaling (PS) and its variations, as well as isotonic regression, beta calibration, and analytical calibration. The data generating process has a mean probability of success of 0.0208. The base model is nearly perfect in expectation and the calibration methods were trained on 100 000 observations (left) and 1 000 000 observations (right). Bins with 10 or less observations were removed.

Since the plots in Fig. 6 provide less information, we relied more on the quantitative assessment to determine how well each calibration method worked in this setting (see Appendix 7). When the base model is nearly perfect in expectation, Platt's scaling with the logit transformation (both with and without the GAM) and beta calibration were generally the most effective calibration methods. Analytical calibration was no longer perfect because of the errors in the base model's predictions, but it also did not perform nearly as poorly as it did when the base model pushed probability estimates towards 0.5 or towards extreme values. When the mean outcome success rate was 0.0022 and the calibration dataset had 100 000 observations, analytical calibration had the lowest mean RMSE and MAE for the ten runs we conducted. However, as the other calibration methods were given more information, either through increasing the size of the calibration dataset or through increasing the mean outcome success rate, the relative effectiveness of analytical calibration diminished. These results again reinforce why Platt's scaling should not be blindly used to calibrate models after undersampling; it often performed considerably worse than the best methods in this setting.

### 4.3 Discussion

Undersampling is a common approach used to address a bias in machine learning models when performing imbalanced classification (e.g., Burmeister et al. 2023; Shin et al. 2023). Other techniques, such as cost-sensitive learning, are also commonly used (e.g., Sun et al. 2007; Araf et al. 2024). While undersampling offers the advantage of faster training times, we do not wish to take a stance on the relative effectiveness of these approaches. Our study has focused on calibration after undersampling as opposed to cost-sensitive learning because we identified that suboptimal practices were being used in this setting.

Our simulation study has shown that the effectiveness of different calibration methods varies substantially based on a number of factors. Although we have considered the level of class imbalance in the data and the

size of the calibration dataset, our primary consideration was the performance of the base model. Our results show that this is an extremely important factor to consider. While our simulation study has shown that traditional Platt's scaling can perform poorly in some cases, we also found that one of the four variations of Platt's scaling was the best-performing calibration method in all cases except with the perfect base model. Thus, Platt's scaling or a variation of it can be a valuable tool for calibrating models after undersampling.

Although traditional Platt's scaling performed poorly when the base model was perfect, the variation of Platt's scaling with the logit transformation performed well. It did not perform as well as analytical calibration because that method perfectly adjusts for undersampling in this setting. However, analytical calibration is risky to use because it can perform terribly with imperfect models. In addition, even with a model that is nearly perfect in expectation, Platt's scaling with the logit transformation was able to outperform it.

Traditional Platt's scaling was not effective when the base model was perfect, but it worked very well when the base model pushed probability estimates towards 0.5 via a sigmoidal relationship. In this setting, the relationship between the model's outputs and the true probabilities is linear on the log odds scale, so the assumptions of the logistic regression model are met. Thus, even though Platt's scaling is unable to properly calibrate a perfect base model after undersampling, it might be suitable sometimes. However, we fit a random forest to the wildland fire data from Section 3 and found that Platt's scaling did a poor job calibrating it, even though the random forest pushed probability estimates towards 0.5 (see Appendix 8 for more details). This suggests that models that push probability estimates towards 0.5 still might not do so in a manner that makes Platt's scaling a good method for calibration after undersampling.

When the base model pushes probability estimates towards 0 or 1, none of the calibration methods worked very well, but Platt's scaling with both the logit transformation and a GAM tended to lead to the best-calibrated probabilities. This is one of the safest calibration choices, as it generally performed fairly well in terms of RMSE and MAE regardless of the base model. Beta calibration also generally performed fairly well, but it struggled relative to Platt's scaling with the logit transformation and a GAM when the base model pushed estimates towards 0 or 1. This was where we found the largest difference in performance between these two methods.

Our simulation study has also shown that the use of Platt's scaling can have undesirable effects on model selection. Consider the case where we fit several base models and would like to compare them so that we can choose a final model. In our simulation study, if we use traditional Platt's scaling to calibrate each model, then we would choose the model that pushes probability estimates towards 0.5 as our final model, even though one of our candidate models is perfect. This could negatively affect our ability to interpret the model as we try to better understand the underlying data generating process.

## 5    Conclusion

We have shown that Platt's scaling is generally not a good choice for calibrating models trained on an undersampled dataset. Although it can work, Platt's scaling relies on the base model having a specific systematic error to properly calibrate the predictions. With a perfect base model, calibration via Platt's scaling results in predictions that underestimate success probabilities for higher probability outcomes. For a field such as wildland fire management, this could cause substantial problems due to underestimation of wildland fires. However, a modified version of Platt's scaling based on the logit of the base model's predictions is an effective calibration approach with a perfect base model. This is also known as beta$[a = b]$ calibration. Using beta calibration (without the restriction that $a = b$) or a logistic GAM instead of a logistic regression model can also lead to improved calibration.

To choose a calibration method in practice, the most robust approach is to compare the different methods and then choose the best one. A practitioner could qualitatively and quantitatively evaluate each method given a particular base model, and then select the best calibration method. For real data, the true probabilities are unknown, so metrics like Brier score or negative logarithmic score would be needed in place of RMSE and MAE. Of the two, we recommend negative logarithmic score because it can better reflect differences in the models (Benedetti 2010). Other metrics may also be viable options, like customized metrics from the Beta family of scoring rules (Merkle & Steyvers 2013). It is also important to note that to obtain an unbiased

estimate of the performance of the entire modelling procedure, an additional dataset is needed. Using the metrics obtained on the testing dataset (which were used to choose the calibration method) will result in a biased estimate.

This quantitative comparison is a relatively time-consuming process, so a practitioner may wish to bypass such a procedure. This may be possible by critically thinking about the base model being used. In general, our results indicate that a practitioner should only use traditional Platt's scaling for calibration after undersampling if it would have been able to calibrate the base model had it been trained on the entire dataset (i.e., without undersampling) (e.g., see Böken 2021). For example, boosted trees and random forests tend to push probability estimates towards 0.5 (e.g., Niculescu-Mizil & Caruana 2005; Guilbert et al. 2024), so one might choose to use Platt's scaling for calibration anyways when using these models. Traditional Platt's scaling can simultaneously account for a base model pushing its estimates towards 0.5 and being miscalibrated due to undersampling, so it might be a good choice in this situation. However, this justification for using Platt's scaling is generally not given in the undersampling literature; oftentimes, the only reasoning given is miscalibration due to undersampling (e.g., Wallace & Dahabreh 2014; Moreau et al. 2020; Peng et al. 2020; Phelps & Woolford 2021a; Burmeister et al. 2023; Shin et al. 2023). Even when models push probability estimates towards 0.5, they cannot be expected to err in a perfectly sigmoidal fashion. As we have seen when using a random forest on the wildland fire data, traditional Platt's scaling still might not work very well in this setting. Consequently, even when using models known to push probability estimates towards 0.5, it still might be better to use another method.

If a practitioner cannot justify using Platt's scaling had they not used undersampling, then we recommend the modification of Platt's scaling with the logit transformation and a GAM. This approach is supported analytically when the base model is perfect, but also provides additional flexibility; the GAM can fit any smooth relationship if given enough training data. We found this flexibility was especially important when the base model pushed probability estimates towards 0 or 1. We expect that it might also be useful in practice, when the errors in the base model might be more complex than those considered in this study (e.g., asymmetrical about 0.5).

### Acknowledgments

We acknowledge the support of the Natural Sciences and Engineering Research Council of Canada (NSERC) through its Postgraduate Scholarship program, Discovery Grant program, and Strategic Networks program.

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

# A    Appendices

## A.1    Appendix 1: Regularization in Platt's scaling

As in the main body of this paper, consider a dataset with a dichotomous response $Y$ that takes value 0 (i.e., the negative class) or 1 (i.e., the positive class). Let $n$ represent the size of the dataset, with $n_0$ and $n_1$ representing the number of negative and positive class observations, respectively. In Platt (1999), the authors use a regularization approach whereby responses are modified such that responses of 0 become $\frac{1}{n_0+2}$ and responses of 1 become $\frac{n_1+1}{n_1+2}$. For large samples such as those considered in our study, this modification would have very little effect.

## A.2    Appendix 2: Reliability plots for calibrated logistic GAM on wildland fire data

**Panel A**

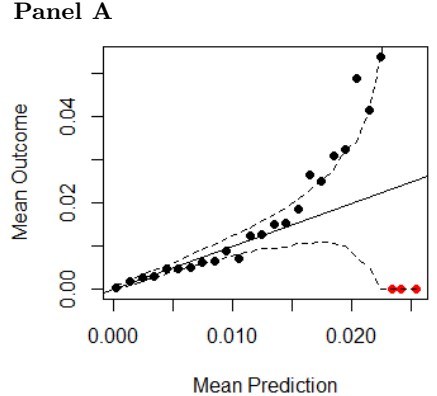

**Panel B**

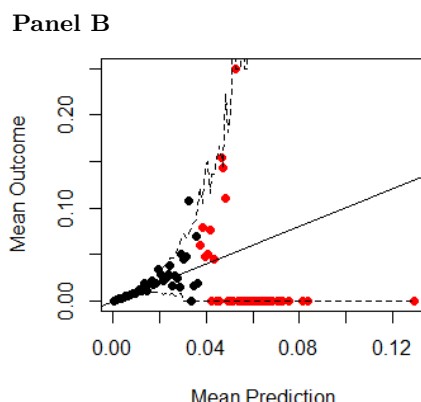

Figure 7: Reliability plots for a logistic Generalized Additive Model (GAM) calibrated using Platt's scaling (Panel A) and Platt's scaling with the logit transformation (Panel B). The solid line is the 45° line and the dashed lines represent 95% prediction intervals, computed assuming the predicted probabilities from the modelling were correct. Red points are those that were removed from the plots in the main body of the paper.

### A.3   Appendix 3: Simulation study set-up details

### A.3.1   Simulating data

For all of the simulated datasets, we generated 10 covariates. Each of these covariates followed a uniform distribution, but with different minimums and maximums. Those values are shown in Table 1.

Table 1: Minimum and maximum values for each of the 10 covariates in the simulated datasets.

| Covariate | Minimum | Maximum |
|-----------|---------|---------|
| 1         | -0.4    | 0.6     |
| 2         | -0.2    | 0.8     |
| 3         | -0.4    | 1.0     |
| 4         | -0.1    | 0.9     |
| 5         | 0.0     | 5.0     |
| 6         | 0.0     | 3.0     |
| 7         | 1.0     | 4.0     |
| 8         | 1.0     | 7.0     |
| 9         | 1.0     | 3.0     |
| 10        | 0.0     | 2.0     |

Based on the 10 covariates, we generated the log odds of success for each observation according to Eq. 11:

$$
\begin{aligned}
\text{logit}(p) = \frac{\log(99)}{40} \big( & x_1 + x_2 + x_3 + x_4 + x_5 + x_6 + x_7 + x_8 + x_9 + x_{10} \\
& + x_1 x_3 + x_2 x_5 + x_4 x_9 + x_6 x_7 + x_8 x_{10} + x_1 x_2 x_3 x_4 + x_1 x_2 x_9 x_{10} \big) - w \log(99)
\end{aligned}
\tag{11}
$$

Here, $w$ is a parameter that can be used to alter the rate at which successes occur. For our simulation study, we set $w$ to 2, 1.5, and 1.1. Undoing the logit operation yields the success probabilities, which were used to simulate the outcomes in the datasets.

### A.3.2 Simulating model outputs

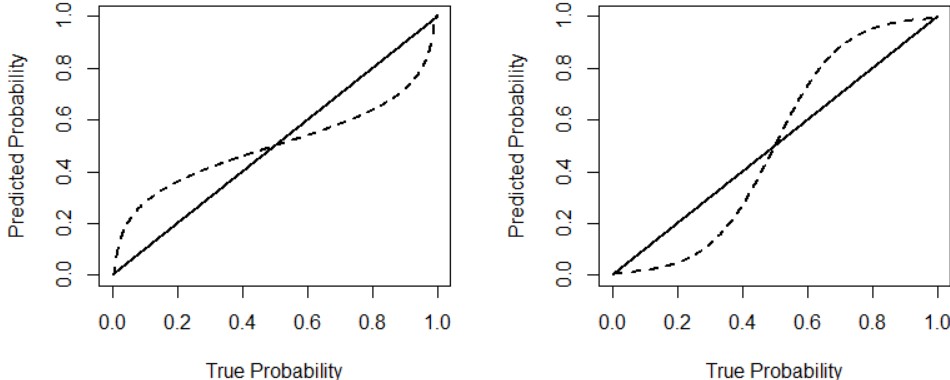

Figure 8: The relationship between the predicted probability and the true probability for a perfect model (solid) and a model with systematic estimation error (dashed). The left plot shows a model that pushes probability estimates towards 0.5 (Eq. 9), while the right plot shows a model that pushes probability estimates towards 0 or 1 (Eq. 10).

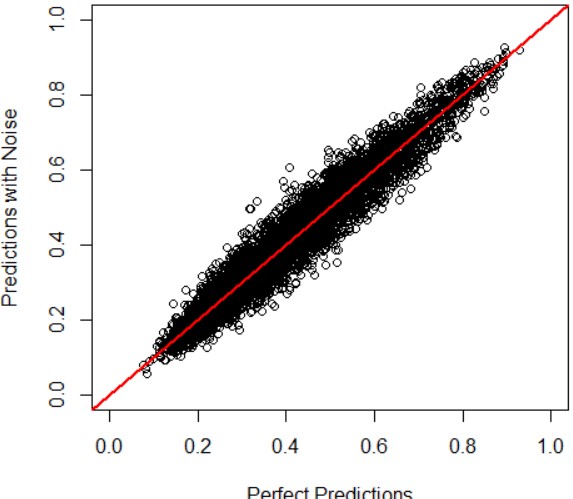

Figure 9: An example of the relationship between the $\hat{\gamma}$'s and the $\gamma$'s when the base model's predictions are altered by a noise variable. The scatterplot shows 5000 observations obtained from the data generating process with a mean outcome probability of 0.0208. The red line is the 45° line.

## A.4 Appendix 4: Results for perfect base model

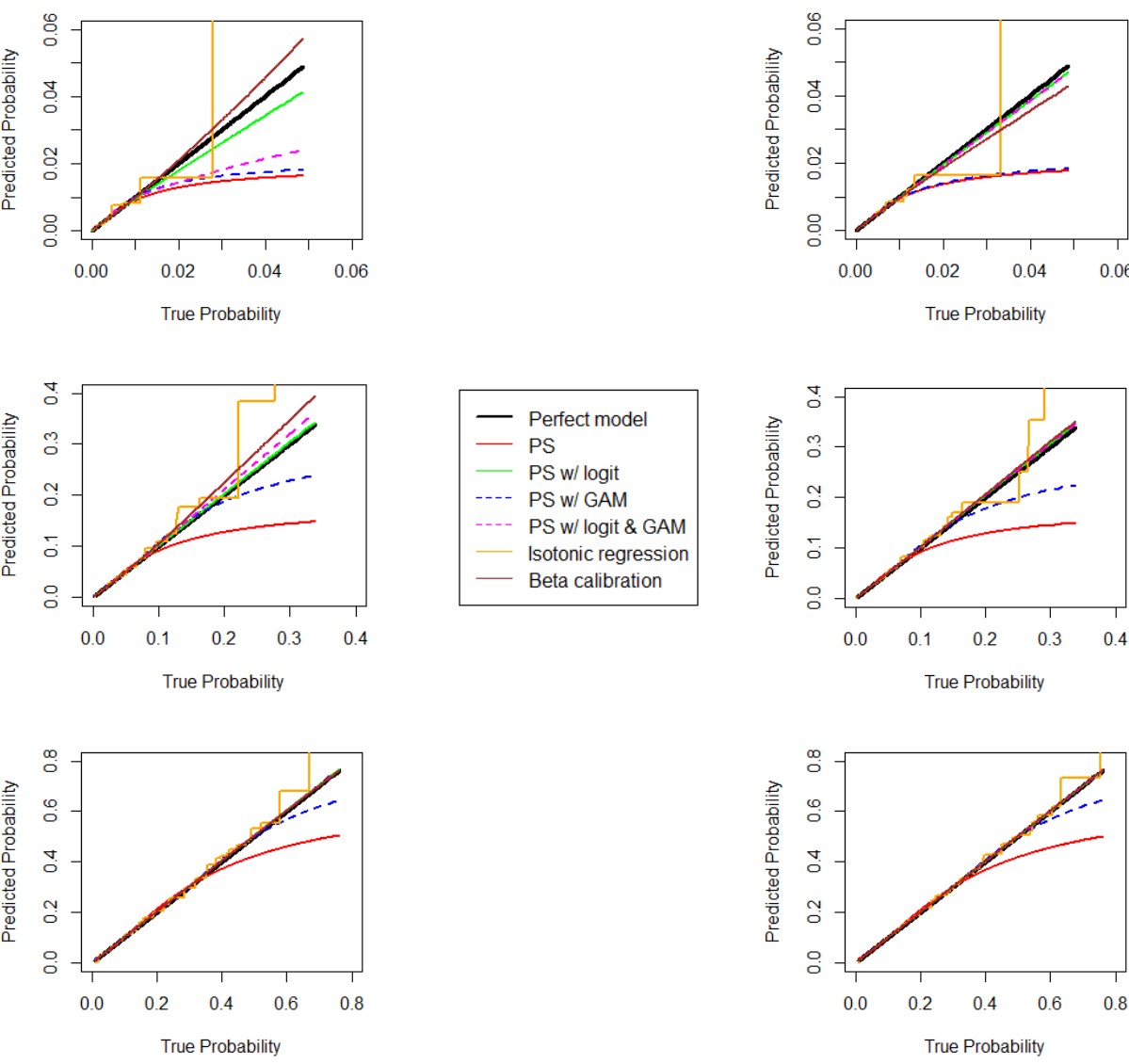

Figure 10: Probability estimates on the testing dataset from Platt's scaling (PS) and its variations, as well as isotonic regression and beta calibration. The data generating processes have a mean probability of success of 0.0022 (top), 0.0208 (middle), and 0.1109 (bottom). The base model is perfect and the calibration methods were trained on 100 000 observations (left) and 1 000 000 observations (right).

Table 2: Root mean squared error (RMSE) and mean absolute error (MAE) for Platt's scaling (PS) and its variations, as well as isotonic regression, beta calibration, and analytical calibration, using 100 000 training observations. The base model used here is perfect. Results are shown for the individual run shown in the figures and for 10 additional runs. For the 10 additional runs, we report the mean and its 95% confidence interval (in parentheses).

| Success rate | Calibration method | RMSE ($\times 10^4$) | | MAE ($\times 10^4$) | |
|---|---|---|---|---|---|
| | | Individual run | Ten runs | Individual run | Ten runs |
| 0.0022 | PS | 4.48 | 4.72 (3.78, 5.66) | 1.12 | 1.65 (1.10, 2.21) |
| | PS w/ logit | 1.77 | 2.35 (1.06, 3.63) | 0.94 | 1.41 (0.77, 2.05) |
| | PS w/ GAM | 4.49 | 4.78 (3.83, 5.73) | 2.69 | 1.65 (1.10, 2.20) |
| | PS w/ logit and GAM | 4.50 | 2.71 (1.33, 4.09) | 2.82 | 1.44 (0.80, 2.09) |
| | Isotonic regression | 8.16 | 10.18 (7.61, 12.75) | 4.24 | 3.65 (3.27, 4.03) |
| | Beta calibration | 1.42 | 3.23 (1.69, 4.76) | 1.12 | 1.54 (0.91, 2.17) |
| | Analytical | 0.00 | 0.00 (0.00, 0.00) | 0.00 | 0.00 (0.00, 0.00) |
| 0.0208 | PS | 31.09 | 31.45 (30.73, 32.17) | 10.83 | 12.04 (10.97, 13.11) |
| | PS w/ logit | 2.05 | 7.84 (4.77, 10.92) | 1.49 | 5.62 (3.61, 7.64) |
| | PS w/ GAM | 12.25 | 17.52 (15.28, 19.77) | 5.37 | 8.02 (6.44, 9.60) |
| | PS w/ logit and GAM | 5.80 | 7.79 (4.69, 10.89) | 2.40 | 5.61 (3.58, 7.64) |
| | Isotonic regression | 42.49 | 36.08 (29.46, 42.69) | 13.58 | 15.92 (14.38, 17.46) |
| | Beta calibration | 12.12 | 8.71 (6.27, 11.14) | 5.24 | 6.01 (4.20, 7.83) |
| | Analytical | 0.00 | 0.00 (0.00, 0.00) | 0.00 | 0.00 (0.00, 0.00) |
| 0.1109 | PS | 87.78 | 87.14 (85.43, 88.84) | 43.88 | 42.79 (40.71, 44.88) |
| | PS w/ logit | 16.97 | 17.91 (14.35, 21.46) | 13.26 | 13.61 (9.74, 17.48) |
| | PS w/ GAM | 25.76 | 34.14 (30.66, 37.63) | 14.93 | 20.52 (17.61, 23.43) |
| | PS w/ logit and GAM | 16.97 | 18.05 (14.47, 21.64) | 13.26 | 13.57 (9.69, 17.45) |
| | Isotonic regression | 65.47 | 73.25 (68.46, 78.05) | 41.76 | 45.33 (43.51, 47.15) |
| | Beta calibration | 16.84 | 19.77 (15.82, 23.71) | 13.27 | 13.71 (9.86, 17.57) |
| | Analytical | 0.00 | 0.00 (0.00, 0.00) | 0.00 | 0.00 (0.00, 0.00) |

Table 3: Root mean squared error (RMSE) and mean absolute error (MAE) for Platt's scaling (PS) and its variations, as well as isotonic regression, beta calibration, and analytical calibration, using 1 000 000 training observations. The base model used here is perfect. Results are shown for the individual run shown in the figures and for 10 additional runs. For the 10 additional runs, we report the mean and its 95% confidence interval (in parentheses).

| Success rate | Calibration method | RMSE ($\times 10^4$) | | MAE ($\times 10^4$) | |
|---|---|---|---|---|---|
| | | Individual run | Ten runs | Individual run | Ten runs |
| | PS | 3.91 | 3.82 (3.67, 3.96) | 1.07 | 1.19 (1.10, 1.29) |
| | PS w/ logit | 0.57 | 0.65 (0.24, 1.05) | 0.26 | 0.43 (0.21, 0.65) |
| | PS w/ GAM | 3.68 | 1.94 (1.66, 2.22) | 1.00 | 0.73 (0.60, 0.87) |
| 0.0022 | PS w/ logit and GAM | 0.58 | 0.69 (0.32, 1.07) | 0.26 | 0.44 (0.23, 0.66) |
| | Isotonic regression | 4.10 | 4.68 (3.51, 5.86) | 1.64 | 1.60 (1.48, 1.72) |
| | Beta calibration | 1.01 | 0.90 (0.58, 1.23) | 0.26 | 0.54 (0.35, 0.72) |
| | Analytical | 0.00 | 0.00 (0.00, 0.00) | 0.00 | 0.00 (0.00, 0.00) |
| | PS | 30.76 | 31.29 (30.84, 31.74) | 11.22 | 10.47 (10.24, 10.70) |
| | PS w/ logit | 3.41 | 1.86 (1.26, 2.46) | 1.57 | 1.34 (0.88, 1.80) |
| | PS w/ GAM | 10.84 | 8.50 (7.70, 9.30) | 3.46 | 2.90 (2.44, 3.37) |
| 0.0208 | PS w/ logit and GAM | 3.41 | 1.98 (1.38, 2.57) | 1.57 | 1.46 (0.93, 1.99) |
| | Isotonic regression | 15.25 | 15.60 (13.21, 18.00) | 6.64 | 7.15 (6.87, 7.43) |
| | Beta calibration | 4.16 | 2.69 (1.71, 3.67) | 1.51 | 1.68 (1.02, 2.35) |
| | Analytical | 0.00 | 0.00 (0.00, 0.00) | 0.00 | 0.00 (0.00, 0.00) |
| | PS | 87.00 | 86.53 (85.66, 87.40) | 40.69 | 39.66 (38.64, 40.67) |
| | PS w/ logit | 5.91 | 5.25 (2.38, 8.13) | 5.06 | 3.73 (2.02, 5.44) |
| | PS w/ GAM | 16.34 | 16.81 (15.14, 18.47) | 6.67 | 7.21 (6.07, 8.35) |
| 0.1109 | PS w/ logit and GAM | 5.91 | 6.40 (3.46, 9.34) | 5.06 | 4.36 (2.63, 6.10) |
| | Isotonic regression | 34.37 | 30.34 (29.07, 31.61) | 20.56 | 19.55 (18.99, 20.11) |
| | Beta calibration | 6.31 | 5.81 (2.92, 8.70) | 5.06 | 4.04 (2.40, 5.68) |
| | Analytical | 0.00 | 0.00 (0.00, 0.00) | 0.00 | 0.00 (0.00, 0.00) |

## A.5 Appendix 5: Results for base model pushing probability estimates towards 0.5

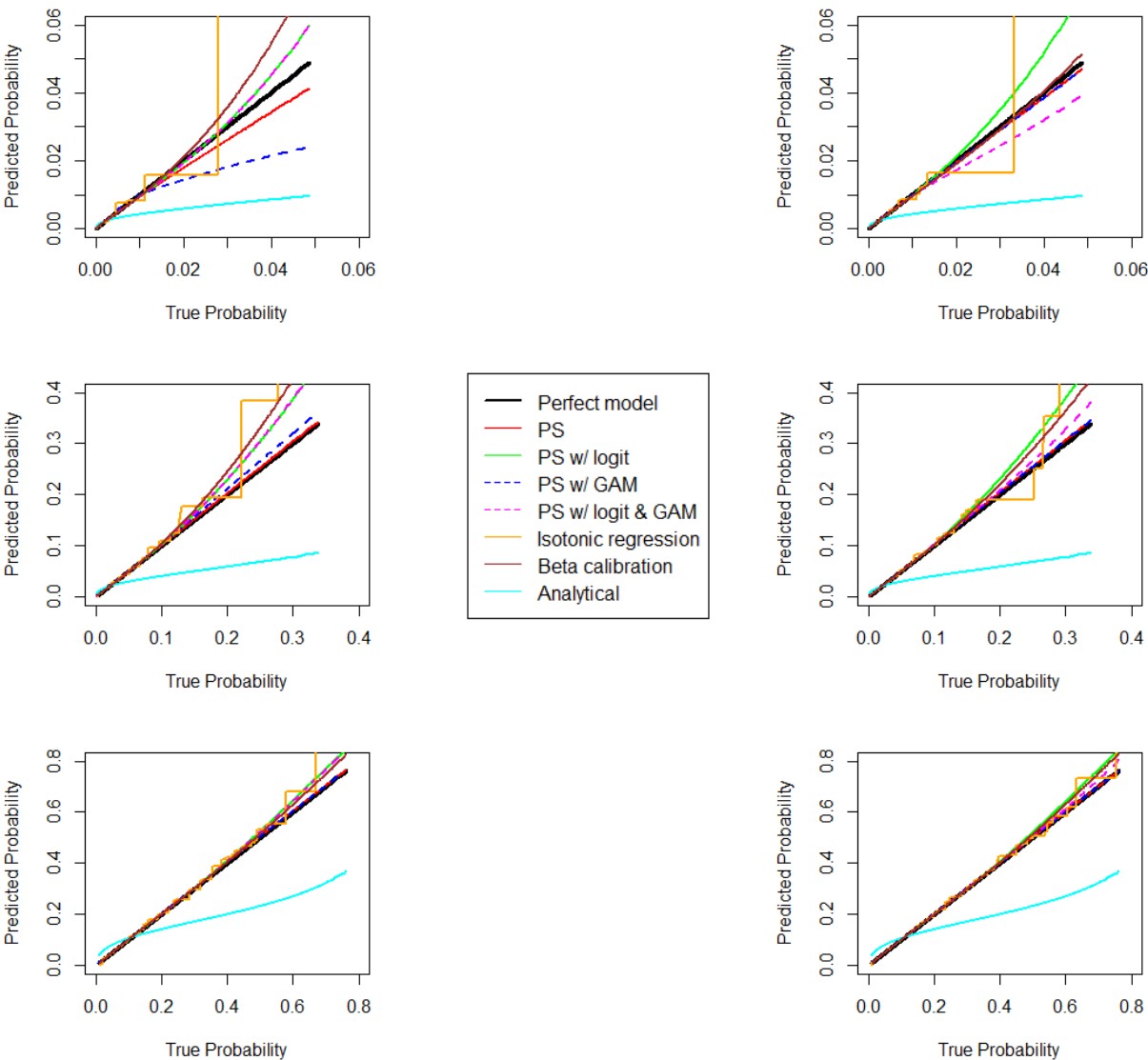

Figure 11: Probability estimates on the testing dataset from Platt's scaling (PS) and its variations, as well as isotonic regression, beta calibration, and analytical calibration. The data generating processes have a mean probability of success of 0.0022 (top), 0.0208 (middle), and 0.1109 (bottom). The base model pushes probability estimates towards 0.5 and the calibration methods were trained on 100 000 observations (left) and 1 000 000 observations (right).

Table 4: Root mean squared error (RMSE) and mean absolute error (MAE) for Platt's scaling (PS) and its variations, as well as isotonic regression, beta calibration, and analytical calibration, using 100 000 training observations. The base model used here pushes probability estimates towards 0.5. Results are shown for the individual run shown in the figures and for 10 additional runs. For the 10 additional runs, we report the mean and its 95% confidence interval (in parentheses).

| Success rate | Calibration method | RMSE ($\times 10^4$) | | MAE ($\times 10^4$) | |
| --- | --- | --- | --- | --- | --- |
| | | Individual run | Ten runs | Individual run | Ten runs |
| | PS | 1.77 | 2.35 (1.06, 3.63) | 0.94 | 1.41 (0.77, 2.05) |
| | PS w/ logit | 1.59 | 2.54 (1.32, 3.77) | 1.08 | 1.48 (0.86, 2.09) |
| | PS w/ GAM | 4.50 | 2.71 (1.33, 4.09) | 2.82 | 1.44 (0.80, 2.09) |
| 0.0022 | PS w/ logit and GAM | 1.59 | 3.47 (2.03, 4.91) | 1.08 | 1.64 (1.01, 2.28) |
| | Isotonic regression | 8.16 | 10.18 (7.61, 12.75) | 4.24 | 3.65 (3.27, 4.03) |
| | Beta calibration | 1.88 | 2.67 (1.41, 3.93) | 1.19 | 1.51 (0.90, 2.11) |
| | Analytical | 14.51 | 14.44 (14.40, 14.47) | 8.18 | 8.16 (8.15, 8.17) |
| | PS | 2.05 | 7.84 (4.77, 10.92) | 1.49 | 5.62 (3.61, 7.64) |
| | PS w/ logit | 11.26 | 13.97 (11.59, 16.34) | 2.51 | 6.07 (4.06, 8.09) |
| | PS w/ GAM | 5.80 | 7.79 (4.69, 10.89) | 2.40 | 5.61 (3.58, 7.64) |
| 0.0208 | PS w/ logit and GAM | 11.53 | 11.56 (8.74, 14.37) | 2.59 | 6.19 (4.30, 8.07) |
| | Isotonic regression | 42.49 | 36.08 (29.46, 42.69) | 13.58 | 15.92 (14.38, 17.46) |
| | Beta calibration | 17.74 | 9.70 (7.18, 12.22) | 4.82 | 6.11 (4.08, 8.15) |
| | Analytical | 130.10 | 129.59 (129.34, 129.84) | 75.24 | 75.07 (74.99, 75.15) |
| | PS | 16.97 | 17.91 (14.35, 21.46) | 13.26 | 13.61 (9.74, 17.48) |
| | PS w/ logit | 27.15 | 28.51 (25.90, 31.12) | 13.39 | 13.77 (9.40, 18.15) |
| | PS w/ GAM | 16.97 | 18.05 (14.47, 21.64) | 13.26 | 13.57 (9.69, 17.45) |
| 0.1109 | PS w/ logit and GAM | 25.47 | 26.02 (22.26, 29.78) | 13.49 | 13.89 (9.52, 18.27) |
| | Isotonic regression | 65.47 | 73.25 (68.46, 78.05) | 41.76 | 45.33 (43.51, 47.15) |
| | Beta calibration | 20.50 | 24.40 (20.43, 28.37) | 14.80 | 14.03 (9.81, 18.26) |
| | Analytical | 506.98 | 505.76 (505.15, 506.37) | 346.29 | 345.70 (345.41, 345.98) |

Table 5: Root mean squared error (RMSE) and mean absolute error (MAE) for Platt's scaling (PS) and its variations, as well as isotonic regression, beta calibration, and analytical calibration, using 1 000 000 training observations. The base model used here pushes probability estimates towards 0.5. Results are shown for the individual run shown in the figures and for 10 additional runs. For the 10 additional runs, we report the mean and its 95% confidence interval (in parentheses).

| Success rate | Calibration method | RMSE ($\times 10^4$) | | MAE ($\times 10^4$) | |
|---|---|---|---|---|---|
| | | Individual run | Ten runs | Individual run | Ten runs |
| | PS | 0.57 | 0.65 (0.24, 1.05) | 0.26 | 0.43 (0.21, 0.65) |
| | PS w/ logit | 1.24 | 1.59 (1.29, 1.89) | 0.44 | 0.53 (0.34, 0.71) |
| | PS w/ GAM | 0.58 | 0.69 (0.32, 1.07) | 0.26 | 0.44 (0.23, 0.66) |
| 0.0022 | PS w/ logit and GAM | 1.69 | 1.30 (0.88, 1.72) | 0.46 | 0.52 (0.34, 0.71) |
| | Isotonic regression | 4.10 | 4.68 (3.51, 5.86) | 1.64 | 1.60 (1.48, 1.72) |
| | Beta calibration | 1.00 | 1.09 (0.76, 1.41) | 0.41 | 0.59 (0.42, 0.76) |
| | Analytical | 14.51 | 14.44 (14.40, 14.47) | 8.18 | 8.16 (8.15, 8.17) |
| | PS | 3.41 | 1.86 (1.26, 2.46) | 1.57 | 1.34 (0.88, 1.80) |
| | PS w/ logit | 11.99 | 10.03 (9.57, 10.48) | 2.01 | 2.81 (2.53, 3.09) |
| | PS w/ GAM | 3.41 | 1.98 (1.38, 2.57) | 1.57 | 1.46 (0.93, 1.99) |
| 0.0208 | PS w/ logit and GAM | 5.62 | 4.20 (3.16, 5.25) | 2.04 | 1.91 (1.37, 2.46) |
| | Isotonic regression | 15.25 | 15.60 (13.21, 18.00) | 6.64 | 7.15 (6.87, 7.43) |
| | Beta calibration | 7.89 | 5.97 (4.68, 7.26) | 1.56 | 2.54 (2.12, 2.96) |
| | Analytical | 130.10 | 129.59 (129.34, 129.84) | 75.24 | 75.07 (74.99, 75.15) |
| | PS | 5.91 | 5.25 (2.38, 8.13) | 5.06 | 3.73 (2.02, 5.44) |
| | PS w/ logit | 21.77 | 21.67 (20.76, 22.58) | 9.06 | 9.71 (8.37, 11.04) |
| | PS w/ GAM | 5.91 | 6.40 (3.46, 9.34) | 5.06 | 4.36 (2.63, 6.10) |
| 0.1109 | PS w/ logit and GAM | 8.03 | 9.70 (7.27, 12.13) | 5.30 | 5.34 (3.78, 6.90) |
| | Isotonic regression | 34.37 | 30.34 (39.07, 31.61) | 20.56 | 19.55 (18.99, 20.11) |
| | Beta calibration | 15.55 | 14.86 (13.42, 16.30) | 8.29 | 8.44 (7.40, 9.47) |
| | Analytical | 506.98 | 505.76 (505.15, 506.37) | 346.29 | 345.70 (345.41, 345.98) |

## A.6 Appendix 6: Results for base model pushing probability estimates towards 0 or 1

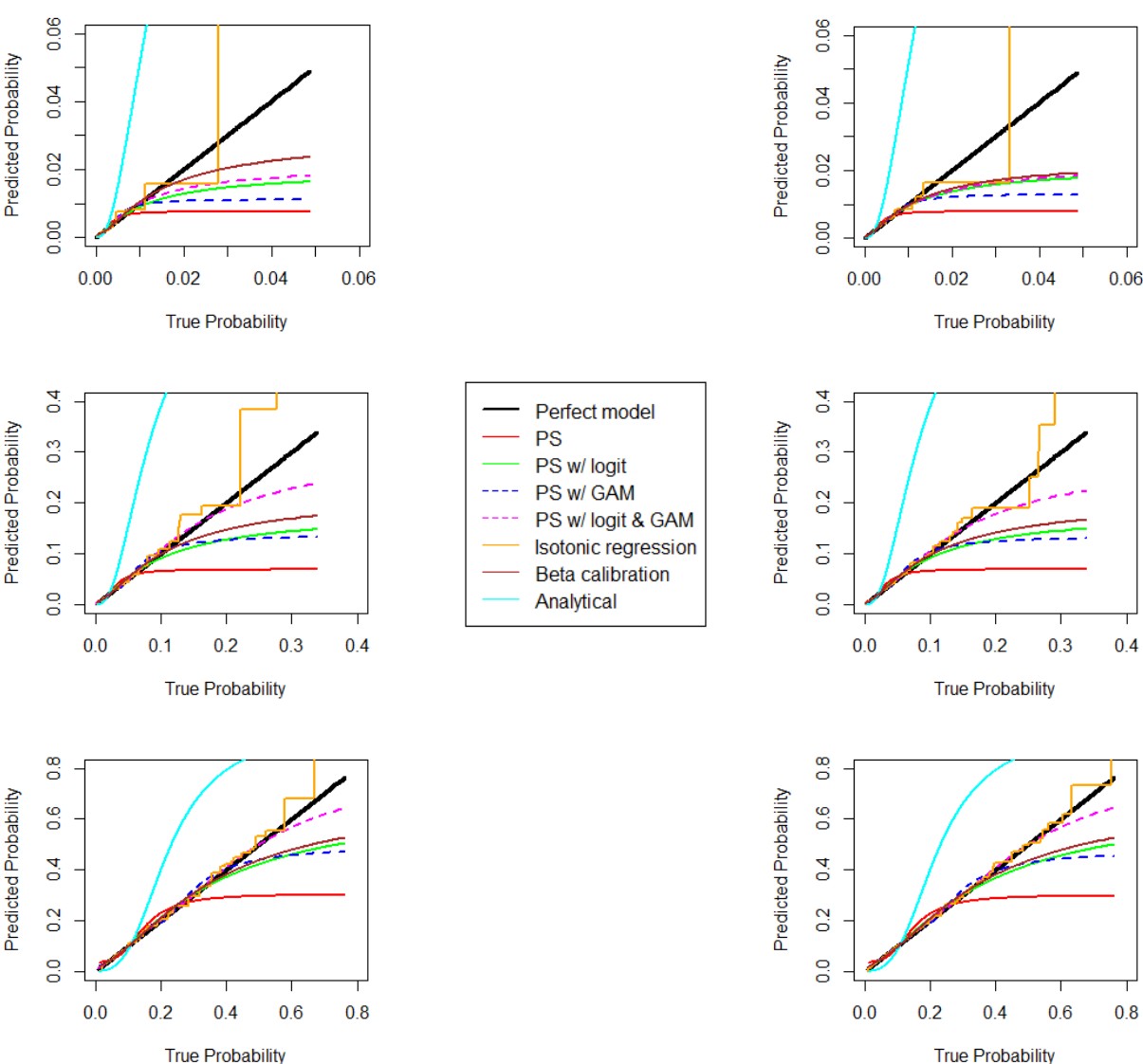

Figure 12: Probability estimates on the testing dataset from Platt's scaling (PS) and its variations, as well as isotonic regression, beta calibration, and analytical calibration. The data generating processes have a mean probability of success of 0.0022 (top), 0.0208 (middle), and 0.1109 (bottom). The base model pushes probabiliy estimates towards 0 or 1 and the calibration methods were trained on 100 000 observations (left) and 1 000 000 observations (right).

Table 6: Root mean squared error (RMSE) and mean absolute error (MAE) for Platt's scaling (PS) and its variations, as well as isotonic regression, beta calibration, and analytical calibration, using 100 000 training observations. The base model used here pushes probability estimates towards 0 or 1. Results are shown for the individual run shown in the figures and for 10 additional runs. For the 10 additional runs, we report the mean and its 95% confidence interval (in parentheses).

| Success rate | Calibration method | RMSE ($\times 10^4$) | | MAE ($\times 10^4$) | |
|---|---|---|---|---|---|
| | | Individual run | Ten runs | Individual run | Ten runs |
| | PS | 8.82 | 8.99 (8.45, 9.53) | 3.27 | 3.48 (3.27, 3.69) |
| | PS w/ logit | 4.48 | 4.72 (3.78, 5.66) | 1.12 | 1.65 (1.10, 2.21) |
| | PS w/ GAM | 6.81 | 6.87 (5.83, 7.90) | 3.64 | 2.75 (2.34, 3.16) |
| 0.0022 | PS w/ logit and GAM | 4.49 | 4.78 (3.83, 5.73) | 2.69 | 1.65 (1.10, 2.20) |
| | Isotonic regression | 8.16 | 10.18 (7.61, 12.75) | 4.24 | 3.65 (3.27, 4.03) |
| | Beta calibration | 2.77 | 4.83 (3.53, 6.12) | 1.72 | 1.73 (1.20, 2.26) |
| | Analytical | 88.13 | 87.69 (87.48, 87.91) | 29.67 | 29.54 (29.47, 29.61) |
| | PS | 75.23 | 74.79 (74.31, 75.28) | 32.16 | 32.97 (31.79, 34.16) |
| | PS w/ logit | 31.09 | 31.45 (30.73, 32.17) | 10.83 | 12.04 (10.97, 13.11) |
| | PS w/ GAM | 32.42 | 34.92 (32.82, 37.03) | 11.74 | 13.82 (12.12, 15.52) |
| 0.0208 | PS w/ logit and GAM | 12.25 | 17.52 (15.28, 19.77) | 5.37 | 8.02 (6.44, 9.60) |
| | Isotonic regression | 42.49 | 36.08 (29.46, 42.69) | 13.58 | 15.92 (14.38, 17.46) |
| | Beta calibration | 22.89 | 27.06 (25.59, 28.54) | 11.34 | 11.41 (10.62, 12.20) |
| | Analytical | 569.12 | 567.35 (566.32, 568.37) | 238.58 | 237.75 (237.31, 238.19) |
| | PS | 248.33 | 247.21 (246.15, 248.28) | 137.80 | 135.35 (133.33, 137.38) |
| | PS w/ logit | 87.78 | 87.14 (85.43, 88.84) | 43.88 | 42.79 (40.71, 44.88) |
| | PS w/ GAM | 84.93 | 85.19 (83.06, 87.31) | 41.13 | 41.92 (38.41, 45.44) |
| 0.1109 | PS w/ logit and GAM | 25.76 | 34.14 (30.66, 37.63) | 14.93 | 20.52 (17.61, 23.43) |
| | Isotonic regression | 65.47 | 73.25 (68.46, 78.05) | 41.76 | 45.33 (43.51, 47.15) |
| | Beta calibration | 75.31 | 73.49 (71.12, 75.86) | 40.00 | 40.59 (38.80, 42.37) |
| | Analytical | 1210.52 | 1208.49 (1207.26, 1209.73) | 735.66 | 734.05 (733.28, 734.82) |

Table 7: Root mean squared error (RMSE) and mean absolute error (MAE) for Platt's scaling (PS) and its variations, as well as isotonic regression, beta calibration, and analytical calibration, using 1 000 000 training observations. The base model used here pushes probability estimates towards 0 or 1. Results are shown for the individual run shown in the figures and for 10 additional runs. For the 10 additional runs, we report the mean and its 95% confidence interval (in parentheses).

| Success rate | Calibration method | RMSE ($\times 10^4$) | | MAE ($\times 10^4$) | |
|---|---|---|---|---|---|
| | | Individual run | Ten runs | Individual run | Ten runs |
| | PS | 8.60 | 8.55 (8.46, 8.65) | 3.44 | 3.47 (3.37, 3.56) |
| | PS w/ logit | 3.91 | 3.82 (3.67, 3.96) | 1.07 | 1.19 (1.10, 1.29) |
| | PS w/ GAM | 4.75 | 4.27 (4.04, 4.50) | 1.35 | 1.37 (1.20, 1.54) |
| 0.0022 | PS w/ logit and GAM | 3.68 | 1.94 (1.66, 2.22) | 1.00 | 0.73 (0.60, 0.87) |
| | Isotonic regression | 4.10 | 4.68 (3.51, 5.86) | 1.64 | 1.60 (1.48, 1.72) |
| | Beta calibration | 3.37 | 3.36 (3.14, 3.58) | 1.01 | 1.14 (1.06, 1.23) |
| | Analytical | 88.13 | 87.69 (87.48, 87.91) | 29.67 | 29.54 (29.47, 29.61) |
| | PS | 75.01 | 74.69 (74.29, 75.09) | 32.61 | 32.10 (31.90, 32.31) |
| | PS w/ logit | 30.76 | 31.29 (30.84, 31.74) | 11.22 | 10.47 (10.24, 10.70) |
| | PS w/ GAM | 32.23 | 31.96 (31.65, 32.28) | 9.88 | 9.52 (9.22, 9.82) |
| 0.0208 | PS w/ logit and GAM | 10.84 | 8.50 (7.70, 9.30) | 3.46 | 2.90 (2.44, 3.37) |
| | Isotonic regression | 15.25 | 15.60 (13.21, 18.00) | 6.64 | 7.15 (6.87, 7.43) |
| | Beta calibration | 24.81 | 25.65 (24.92, 26.38) | 10.35 | 9.65 (9.43, 9.86) |
| | Analytical | 569.12 | 567.35 (566.32, 568.37) | 238.58 | 237.75 (237.31, 238.19) |
| | PS | 248.37 | 247.10 (246.26, 247.94) | 134.63 | 133.62 (132.79, 134.46) |
| | PS w/ logit | 87.00 | 86.53 (85.66, 87.40) | 40.69 | 39.66 (38.64, 40.67) |
| | PS w/ GAM | 84.33 | 84.25 (83.52, 84.97) | 33.53 | 32.49 (31.84, 33.13) |
| 0.1109 | PS w/ logit and GAM | 16.34 | 16.81 (15.14, 18.47) | 6.67 | 7.21 (6.07, 8.35) |
| | Isotonic regression | 34.37 | 30.34 (29.07, 31.61) | 20.56 | 19.55 (18.99, 20.11) |
| | Beta calibration | 71.92 | 72.44 (71.49, 73.38) | 37.90 | 37.28 (36.46, 38.10) |
| | Analytical | 1210.52 | 1208.49 (1207.26, 1209.73) | 735.66 | 734.05 (733.28, 734.82) |

## A.7   Appendix 7: Results for base model that is nearly perfect in expectation

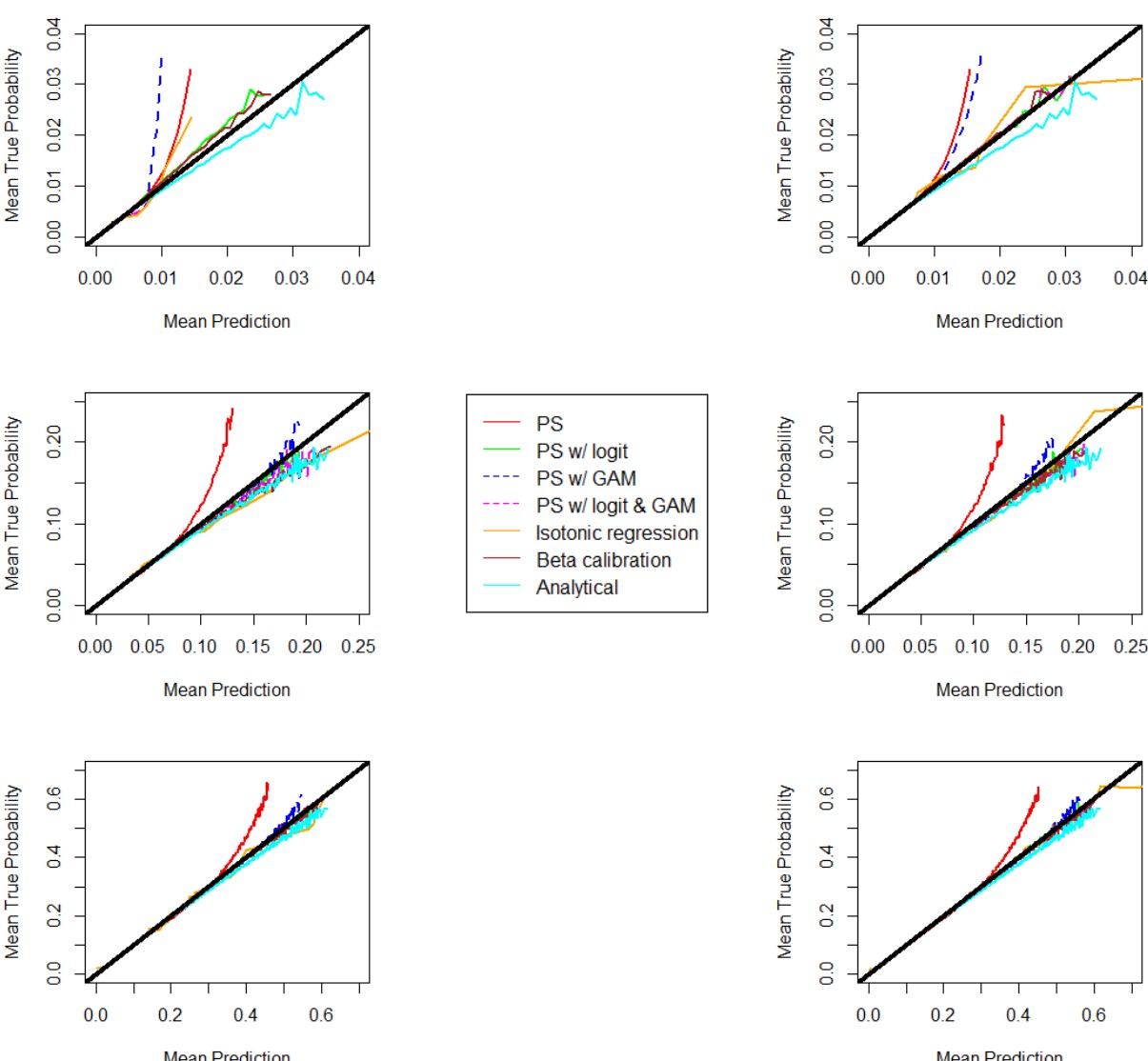

Figure 13: Reliability plots for Platt's scaling (PS) and its variations, as well as isotonic regression, beta calibration, and analytical calibration. The data generating processes have a mean probability of success of 0.0022 (top), 0.0208 (middle), and 0.1109 (bottom). The base model is nearly perfect in expectation and the calibration methods were trained on 100 000 observations (left) and 1 000 000 observations (right). Bins with 10 or less observations were removed.

Table 8: Root mean squared error (RMSE) and mean absolute error (MAE) for Platt's scaling (PS) and its variations, as well as isotonic regression, beta calibration, and analytical calibration, using 100 000 training observations. The base model used here is nearly perfect in expectation. Results are shown for the individual run shown in the figures and for 10 additional runs. For the 10 additional runs, we report the mean and its 95% confidence interval (in parentheses).

| Success rate | Calibration method | RMSE ($\times 10^4$) | | MAE ($\times 10^4$) | |
|---|---|---|---|---|---|
| | | Individual run | Ten runs | Individual run | Ten runs |
| | PS | 7.16 | 7.45 (6.74, 8.15) | 3.64 | 3.79 (3.56, 4.02) |
| | PS w/ logit | 6.08 | 6.40 (5.69, 7.10) | 3.53 | 3.64 (3.39, 3.88) |
| | PS w/ GAM | 9.94 | 7.34 (6.61, 8.07) | 5.33 | 3.82 (3.51, 4.12) |
| 0.0022 | PS w/ logit and GAM | 9.62 | 6.45 (5.65, 7.26) | 5.02 | 3.67 (3.36, 3.98) |
| | Isotonic regression | 9.97 | 10.24 (8.88, 11.61) | 5.82 | 5.05 (4.72, 5.38) |
| | Beta calibration | 6.02 | 6.67 (5.75, 7.59) | 3.53 | 3.71 (3.46, 3.95) |
| | Analytical | 6.20 | 6.16 (6.15, 6.18) | 3.50 | 3.49 (3.49, 3.50) |
| | PS | 58.88 | 58.94 (58.53, 59.35) | 33.05 | 33.44 (32.91, 33.96) |
| | PS w/ logit | 50.95 | 51.46 (50.89, 52.04) | 31.08 | 31.49 (31.03, 31.94) |
| | PS w/ GAM | 52.34 | 53.62 (52.84, 54.39) | 31.48 | 32.09 (31.58, 32.60) |
| 0.0208 | PS w/ logit and GAM | 51.40 | 51.50 (50.95, 52.05) | 31.12 | 31.50 (31.05, 31.95) |
| | Isotonic regression | 64.23 | 63.76 (60.03, 67.48) | 34.61 | 35.31 (34.63, 35.99) |
| | Beta calibration | 52.54 | 51.67 (51.09, 52.26) | 31.39 | 31.58 (31.15, 32.02) |
| | Analytical | 54.06 | 53.88 (53.77, 53.98) | 32.39 | 32.33 (32.29, 32.37) |
| | PS | 223.06 | 222.43 (221.63, 223.23) | 151.10 | 150.20 (149.37, 151.04) |
| | PS w/ logit | 205.99 | 205.72 (205.29, 206.15) | 144.19 | 143.56 (142.84, 144.29) |
| | PS w/ GAM | 207.14 | 207.40 (206.90, 207.91) | 144.72 | 144.34 (143.56, 145.11) |
| 0.1109 | PS w/ logit and GAM | 205.99 | 205.80 (205.40, 206.19) | 144.18 | 143.55 (142.82, 144.28) |
| | Isotonic regression | 215.20 | 217.77 (216.56, 218.97) | 150.11 | 150.35 (149.51, 151.18) |
| | Beta calibration | 205.99 | 206.07 (205.66, 206.47) | 144.16 | 143.56 (142.82, 144.31) |
| | Analytical | 213.50 | 213.18 (212.94, 213.42) | 147.77 | 147.59 (147.44, 147.75) |

Table 9: Root mean squared error (RMSE) and mean absolute error (MAE) for Platt's scaling (PS) and its variations, as well as isotonic regression, beta calibration, and analytical calibration, using 1 000 000 training observations. The base model used here is nearly perfect in expectation. Results are shown for the individual run shown in the figures and for 10 additional runs. For the 10 additional runs, we report the mean and its 95% confidence interval (in parentheses).

| Success rate | Calibration method | RMSE ($\times 10^4$) | | MAE ($\times 10^4$) | |
|---|---|---|---|---|---|
| | | Individual run | Ten runs | Individual run | Ten runs |
| | PS | 6.84 | 6.80 (6.70, 6.90) | 3.54 | 3.57 (3.53, 3.60) |
| | PS w/ logit | 5.79 | 5.79 (5.72, 5.85) | 3.34 | 3.37 (3.35, 3.40) |
| | PS w/ GAM | 6.57 | 6.02 (5.89, 6.15) | 3.49 | 3.42 (3.39, 3.46) |
| 0.0022 | PS w/ logit and GAM | 5.79 | 5.79 (5.73, 5.85) | 3.34 | 3.37 (3.35, 3.40) |
| | Isotonic regression | 6.89 | 7.33 (6.68, 7.98) | 3.72 | 3.75 (3.68, 3.82) |
| | Beta calibration | 5.80 | 5.83 (5.77, 5.88) | 3.34 | 3.38 (3.35, 3.42) |
| | Analytical | 6.20 | 6.16 (6.15, 6.18) | 3.50 | 3.49 (3.49, 3.50) |
| | PS | 58.69 | 58.90 (58.59, 59.20) | 33.21 | 32.98 (32.92, 33.04) |
| | PS w/ logit | 51.04 | 50.72 (50.62, 50.82) | 31.20 | 31.08 (31.02, 31.13) |
| | PS w/ GAM | 51.93 | 51.44 (51.15, 51.74) | 31.36 | 31.21 (31.15, 31.28) |
| 0.0208 | PS w/ logit and GAM | 51.16 | 50.74 (50.60, 50.88) | 31.19 | 31.08 (31.02, 31.13) |
| | Isotonic regression | 53.55 | 53.31 (52.05, 54.58) | 32.11 | 31.93 (31.86, 32.00) |
| | Beta calibration | 51.35 | 50.79 (50.63, 50.94) | 31.20 | 31.09 (31.03, 31.16) |
| | Analytical | 54.06 | 53.88 (53.77, 53.98) | 32.39 | 32.33 (32.29, 32.37) |
| | PS | 222.87 | 222.31 (221.89, 222.73) | 150.24 | 149.52 (149.40, 149.64) |
| | PS w/ logit | 205.43 | 205.08 (204.83, 205.33) | 143.47 | 142.97 (142.86, 143.07) |
| | PS w/ GAM | 205.93 | 205.62 (205.38, 205.87) | 143.51 | 143.06 (142.95, 143.17) |
| 0.1109 | PS w/ logit and GAM | 205.40 | 205.05 (204.79, 205.31) | 143.38 | 142.87 (142.73, 143.00) |
| | Isotonic regression | 207.59 | 207.41 (207.16, 207.67) | 144.62 | 144.26 (144.09, 144.42) |
| | Beta calibration | 205.47 | 205.10 (204.86, 205.35) | 143.39 | 142.90 (142.80, 143.01) |
| | Analytical | 213.50 | 213.18 (212.94, 213.42) | 147.77 | 147.59 (147.44, 147.75) |

## A.8 Appendix 8: Results for random forest fit to wildland fire data

In the main body of the paper, we present results from using Platt's scaling and Platt's scaling with the logit transformation to calibrate a logistic GAM trained on wildland fire data. Here, we do the same but with a random forest as our base model. To obtain a nearly balanced training dataset, we used a sampling rate of 0.08%. To assess the calibration of the random forest on the data generating process it was trained on, we constructed a testing dataset using the same sampling rate. We then created a reliability plot using bin widths of 0.01. We also fit a GAM with weights based on the number of observations in each bin and overlaid the model on the reliability plot. The resulting plot is shown in Fig. 14. Based on the fit of the GAM, the random forest appears to slightly push probability estimates towards 0.5, showing agreement with Niculescu-Mizil & Caruana (2005).

Despite the random forest pushing probability estimates towards 0.5, calibrating the model using Platt's scaling did not seem to work very well. Like with a logistic GAM as the base model, using Platt's scaling leads to underestimating the probability of a fire occurrence for the higher probability cases (see Panel A in Fig. 15). When using Platt's scaling with the logit transformation, this systematic error is eliminated. However, although most points fall within the 95% prediction intervals, it does appear that there might be a slight bias towards overestimating the higher probability cases. This might be due to the miscalibration observed in Fig. 14. Given that the situation where the base model pushes probability estimates towards 0.5 is the one in which it seemed Platt's scaling could be effective, these results suggest an extreme amount of caution should be taken whenever choosing to use Platt's scaling for calibration after undersampling.

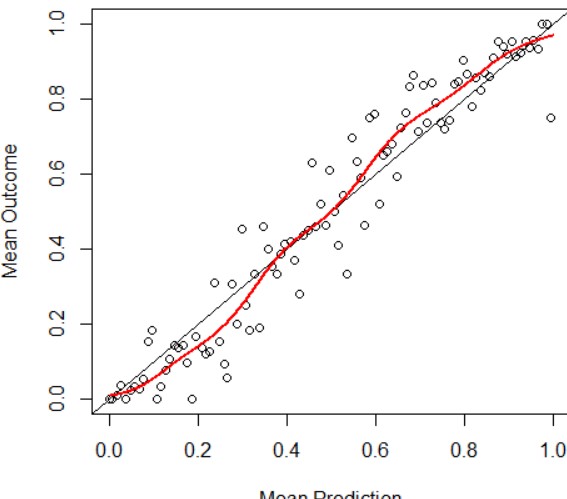

Figure 14: A reliability plot for a random forest with a testing dataset created using undersampling (i.e., the same process used to create the training dataset). The black line is the 45° line and the red line represents a Generalized Additive Model (GAM) fit to the data used to create the reliability plot (i.e., bins of mean predictions and mean outcomes), with weights based on the number of observations in each bin.

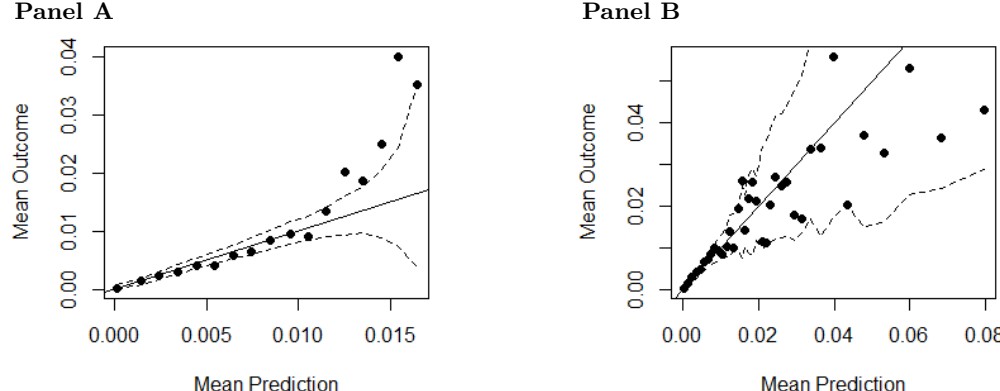

Figure 15: Reliability plots for a random forest calibrated using Platt's scaling (Panel A) and Platt's scaling with the logit transformation (Panel B). The solid line is the 45° line and the dashed lines represent 95% prediction intervals, computed assuming the predicted probabilities from the modelling were correct. Bins with 50 or less observations were removed.

