# OpenReview forum: "Using Platt’s scaling for calibration after undersampling — limitations and how to address them"
_TMLR — Accepted by TMLR_

### Review · Reviewer_qWR1 · 2025-06-26

**Summary Of Contributions:**

This paper studies an interesting and practically relevant problem: In imbalanced binary classification problems, it is often useful to downsample the majority class in the training set when training the model. How can we then ensure the model is calibrated?

The main analytical findings are not difficult to prove but provide practically useful insights. First, the authors show that if the base classifier is perfect on the undersampled training set, then applying Platt’s scaling to this model cannot achieve calibration on the true distribution because the functional form of the true calibration function lies outside the logistic family that Platt scaling assumes. This result also leads naturally to an adaption of Platt’s scaling that does allow for perfect calibration: simply take the logit of the model’s output before applying Platt scaling. Second, the authors show that if the base model is not perfect but rather biases towards outputting probabilities close to 0.5 (specifically, if the relationship between the true and predicted probabilities is sigmoidal), then Platt’s scaling will be able to achieve perfect calibration. For imperfect models that bias towards outputting extreme probabilities, no analytical results are provided but the authors suggest running Platt’s scaling with a GAM to allow for sufficient flexibility (the downside is that more calibration data is required). The analytical findings are then confirmed via simulations using several data generating processes.

**Audience:**

Yes

**Broader Impact Concerns:**

None.

**Claims And Evidence:**

Yes

**Requested Changes:**

Below are a few suggestions and points of confusion that would be useful to clarify.

Sec 3:
* Not sure if I missed this, but what is the number of calibration examples? This would be good to mention.
* “We also added 95% prediction intervals, computed assuming the predicted probabilities were correct.” — what exactly does this mean?
* Suggestions for Figure 1: (1) use wider bins to reduce noise and avoid throwing bins out. (2) Make the axes the same for the two plots to make comparison easier.
* Figure 2 is confusing. Is “offset” on the y-axis referring to log(pi_0) as described in the first paragraph of p. 6? Why is that changing?

Sec 4:
* On page 8 it says “using the GAM offered worse performance because of its additional flexibility” — why doesn’t logit + GAM suffer from this flexibility issue?
* The yellow line for Analytical calibration is quite hard to see in the plots. Perhaps this can be replaced with another color, such as cyan.
* Increasing the line width may improve the readability of the plots

General questions: I would appreciate an answer to these questions here but it is not necessary to incorporate responses to these questions into the paper, although feel free to if you think it will be useful to the intended audience.
1. What happens if you directly train a classifier on a representative dataset? How does the calibration compare to the two-stage approach considered in this paper? (Perhaps this is covered in the cited related work)
2. How is the accuracy of the classifier affection by calibration? e.g., suppose whenever the classifier outputs a probability greater than 0.5, we predict Y=1. Does improving calibration ever hurt accuracy?

**Strengths And Weaknesses:**

Strengths:
* The paper is well written.
* The key message is demonstrated both analytically and empirically, including on real wildfire occurrence data.
* The conclusion section is great. It provides useful guidance to practitioners who want to calibrate binary classification models trained on datasets in which the majority class is undersampled.

Weaknesses: (see Requested Changes for details)
* Figures could be improved.
* Presentation of Sec 3 could be improved.

---

> ### Author Response · Authors · 2025-07-12
> **Response to Reviewer qWR1 (Weaknesses and Requested Changes)**
>
> Thank you for your helpful and thorough feedback on our paper. We have addressed individual comments below.
>
> Comment: Not sure if I missed this, but what is the number of calibration examples? This would be good to mention.
>
> Response: You’re right, we had not included this detail in the paper. There are 2 262 017 observations in the calibration dataset for the wildfire example. We will add this when we make revisions.
>
> Comment: “We also added 95% prediction intervals, computed assuming the predicted probabilities were correct.” — what exactly does this mean?
>
> Response: These are pointwise intervals that were computed by simulating outcomes for each bin as if the model’s predictions were correct. The intervals correspond to the 2.5 and 97.5 quantiles from simulated outcomes under this assumption. We will provide these details in a revision.
>
> Comment: Suggestions for Figure 1: (1) use wider bins to reduce noise and avoid throwing bins out. (2) Make the axes the same for the two plots to make comparison easier.
>
> Response: We had explored different bin widths prior to submission and found that there is a trade-off here between reducing noise and losing information. For example, a model that is not locally well-calibrated might appear well-calibrated with sufficiently wide bins. We have tried slightly widening the bins to 0.0015 (instead of 0.001), with the resulting plots shown below. For the plot with traditional Platt’s scaling, this worked fine. For the plot with the logit transformation, we still had to drop some bins. We used a threshold of having more than 25 observations (instead of 50 before), leading to the exclusion of results from 191 cases (instead of 362 before). Another option is to increase the bin size for larger probabilities, but this can also lead to odd looking plots. Considering these options, do you see a clear best choice that you would like implemented in the paper?
>
> With regards to the axes of the plots, these plots are really standalone figures. It’s the shape that’s important to consider, reflecting the calibration of each model, not the actual values along the axes. We think it might be better for us to clearly indicate that these plots are not directly comparable by making them separate figures in the paper or at least two panels within the same figure. Our concern with making the axes the same is that it might encourage a comparison between the plots that we do not want the reader to make.
>
> Comment: Figure 2 is confusing. Is “offset” on the y-axis referring to log(pi_0) as described in the first paragraph of p. 6? Why is that changing?
>
> Response:  “Offset” refers to the predictions obtained from the method using the offset to account for the undersampling. We will change this to “Predictions from Model with Offset” to make this clearer. You are correct that the offset itself is not changing.
>
> Comment: On page 8 it says “using the GAM offered worse performance because of its additional flexibility” — why doesn’t logit + GAM suffer from this flexibility issue?
>
> Response: This sentence is actually referring to the model with the logit and GAM. It is comparing between the models with and without the GAM, but both using the logit transformation. We will clarify this sentence in a revision.
>
> Comment: The yellow line for Analytical calibration is quite hard to see in the plots. Perhaps this can be replaced with another color, such as cyan.
>
> Response: Thank you for pointing this out. We will use cyan when we revise the paper.
>
> Comment: Increasing the line width may improve the readability of the plots.
>
> Response: We will increase the line widths in a revision.

---

> > ### Author Response · Authors · 2025-07-12
> > **Response to Reviewer qWR1 (General Questions)**
> >
> > Comment: What happens if you directly train a classifier on a representative dataset? How does the calibration compare to the two-stage approach considered in this paper? (Perhaps this is covered in the cited related work)
> >
> > Response: This is an interesting question, and the answer depends on the classifier. It has been shown that logistic regression is biased towards the majority class (King and Zheng, 2021), but this is a small sample bias and would have a limited effect for the large datasets considered in our paper. For machine learning classifiers, there is a widespread belief that models are biased towards the majority class. For example, see Guo et al. (2008), Leevy et al. (2018), or Megahed et al. (2021). However, some of our recent work has suggested that decision trees can actually be biased towards the minority class. (We will not point you to this work at this time because it would remove the anonymity from the review process, but we are happy to point you to it after the review process is completed if you’d like.)
> >
> > Guo, X., Yin, Y., Dong, C., Yang, G., & Zhou, G. (2008). On the class imbalance problem. In 2008 Fourth International Conference on Natural Computation (Vol. 4, pp. 192-201). IEEE.
> >
> > King, G., & Zeng, L. (2001). Logistic regression in rare events data. Political Analysis, 9(2),137-163.
> >
> > Leevy, J. L., Khoshgoftaar, T. M., Bauder, R. A., & Seliya, N. (2018). A survey on addressing high-class imbalance in big data. Journal of Big Data, 5(1), 1-30.
> >
> > Megahed, F. M., Chen, Y. J., Megahed, A., Ong, Y., Altman, N., & Krzywinski, M. (2021). The class imbalance problem. Nat Methods, 18(11), 1270-1272.
> >
> > Comment: How is the accuracy of the classifier affection by calibration? e.g., suppose whenever the classifier outputs a probability greater than 0.5, we predict Y=1. Does improving calibration ever hurt accuracy?
> >
> > Response: We don’t believe that post-hoc calibration as we have done should ever systematically hurt accuracy. If anything, improving calibration should improve accuracy. For example, consider an extremely imbalanced data generating process. If we use undersampling to generate the training dataset, the model’s predictions will be inflated. This will result in more predictions greater than 0.5 (and therefore predictions of Y=1) than there should be. In this setting, it’s entirely possible that we would obtain a model whose accuracy is worse than the baseline accuracy obtained from predicting Y=0 every time. By calibrating the model, the predicted probabilities will drop in magnitude and we will obtain many more Y=0 predictions. If the base model has learned meaningful relationships, we should now be able to improve upon the baseline accuracy.

---

> > ### Comment · Reviewer_qWR1 · 2025-07-15
> > **Reply to Author**
> >
> > Thank you to the authors for the response. I am satisfied with most of the answers but have some question remaining about the following points.
> >
> > > Comment: “We also added 95% prediction intervals, computed assuming the predicted probabilities were correct.” — what exactly does this mean?
> >
> > > Response: These are pointwise intervals that were computed by simulating outcomes for each bin as if the model’s predictions were correct. The intervals correspond to the 2.5 and 97.5 quantiles from simulated outcomes under this assumption. We will provide these details in a revision.
> >
> > Do you do something like this? If there are $n_p$ forecasts that are (close to) probability $p$, then the confidence interval is given by the 2.5 and 97.5 quantiles of Binomial($n_p$, $p$).
> >
> > > with the resulting plots shown below.
> >
> > I don't think I can see the plot. If I need to click somewhere to see it, please let me know.
> >
> > > Our concern with making the axes the same is that it might encourage a comparison between the plots that we do not want the reader to make.
> >
> > I see. I am guessing this due to you having to throw bins out? If no bins are thrown out, then I do think it would be meaningful to be able to make the comparison, because we observe that applying the logit transformation yields higher predicted probabilities.
> >
> > > Comment: On page 8 it says “using the GAM offered worse performance because of its additional flexibility” — why doesn’t logit + GAM suffer from this flexibility issue?
> >
> > > Response: This sentence is actually referring to the model with the logit and GAM. It is comparing between the models with and without the GAM, but both using the logit transformation. We will clarify this sentence in a revision.
> >
> > I was actually referring to how the pink lines (logit+GAM) in Fig 3 are pretty well calibrated, despite, to my understanding, having the same amount of flexibility as the blue lines (GAM only). Do you have a hypothesis as to why? I apologize for not being more clear in my initial comment.

---

> > > ### Author Response · Authors · 2025-07-16
> > > **Response to Reviewer qWR1**
> > >
> > > Thank you for your prompt response to our comments.
> > >
> > > > Do you do something like this? If there are $n_p$ forecasts that are (close to) probability $p$, then the confidence interval is given by the 2.5 and 97.5 quantiles of Binomial($n_p$, $p$).
> > >
> > > This is close, but not quite what we do. If there are $n_p$ forecasts within a bin, then the interval is given by the 2.5 and 97.5 quantiles of the sum of $n_p$ Bernoulli trials with the probability for each trial corresponding to the forecast.
> > >
> > > > I don't think I can see the plot. If I need to click somewhere to see it, please let me know.
> > >
> > > Sorry, this statement was included in error. We realized that we could not include images in our response but missed removing this from our response.
> > >
> > > > I see. I am guessing this due to you having to throw bins out? If no bins are thrown out, then I do think it would be meaningful to be able to make the comparison, because we observe that applying the logit transformation yields higher predicted probabilities.
> > >
> > > You're right that throwing bins out makes it even harder to make the comparison, but this is actually not why we said this. Our reasoning is that these plots are designed to assess the calibration of the models, not the magnitude of the predictions from each model. Even though we observe that traditional Platt's scaling yields lower predicted probabilities, it is possible for a model to exhibit this behavior while still being well-calibrated. Consider this very simple case as an example. If we have a data generating process where 50% of cases have a probability of 0.01 and 50% of cases have a probability of 0.03, a model that predicts 0.02 for all cases is well-calibrated. This is true even though it underestimates the probability for every case with a true probability of 0.03. We agree with you that it is helpful to be able to observe that applying the logit transformation yields higher predicted probabilities, but we believe Fig. 2 is better for making this observation than Fig. 1.
> > >
> > > > I was actually referring to how the pink lines (logit+GAM) in Fig 3 are pretty well calibrated, despite, to my understanding, having the same amount of flexibility as the blue lines (GAM only). Do you have a hypothesis as to why? I apologize for not being more clear in my initial comment.
> > >
> > > Even with the flexibility from the GAM, there is an assumption in place about the relationship between the model predictions and the true probabilities. For the logit+GAM, the assumption about the relationship is correct for the base model considered in Fig. 3. For the GAM only model, the assumption about the relationship is incorrect. We can see that the GAM only model is "fighting against" its incorrect assumption, as the fit of this model (blue line) is much better than the fit of traditional Platt's scaling (red line), but it is not enough to make the GAM only model perform like the logit+GAM model.

---

### Review · Reviewer_qpox · 2025-06-30

**Summary Of Contributions:**

In this paper, the authors examine how Platt’s scaling is used to calibrate models trained on undersampled, imbalanced data and highlight why it often fails in this context. To address the problem, they propose straightforward fixes, such as applying a logit transformation to model predictions before using Platt’s scaling or opting for more flexible methods like beta calibration and logistic GAMs. They have done a detailed simulation study that evaluates these methods in real-world error situations, demonstrating that the improved techniques greatly enhance calibration compared to the basic Platt’s scaling. The paper also offers practical guidance for choosing calibration methods after undersampling, warning against using Platt’s scaling by default without checking its assumptions.

**Audience:**

Yes

**Claims And Evidence:**

Yes

**Requested Changes:**

For the detailed comments, see Strength and Weakness Section. Here is the list of requested changes:

**[Scope Clarification]** I am concerned if it would be better to explain the whole story as a comprehensive analysis of different calibration for undersampling-induced bias directly, and I think it is at least necessary to justify why undersampling plus post-hoc calibration is preferable compared to alternative strategies or why it is somehow important in the introduction.

**[Add more model base]** At least at the random forest, boosted trees, and naive bayes instead of just the logisitc GAM to show that the assumption that the author have made for the statement of probability distribution is hold to be true in the paper.

**[More Real World Dataset Evaluation]** Try to add more unbalance real-world dataset to strengthen the paper.

**[Add Error bars]**

**Strengths And Weaknesses:**

## Strength
**[Comprehensive Theoretical Analysis]** The paper provides a clear and well-structured proof of why Platt’s scaling fails to correct undersampling bias without additional transformation, making the theoretical limitation explicit and easy to follow.

**[Clear Motivation Examples]** The authors motivate the work with a concrete example that highlights real modeling challenges in imbalanced classification settings, and they offer actionable recommendations for improving calibration in such contexts.

**[Comprehensive Simulation Settings]** The simulation study is carefully designed to mimic realistic modeling scenarios, including cases where probability estimates are pushed toward 0.5 (as in ensembles) or toward extreme values (as in Naive Bayes), making the evaluation comprehensive and practically relevant.

## Weakness

**[Limited Motivation]** While the paper clearly demonstrates that Platt’s scaling fails to correct undersampling-induced bias, the proposed alternative methods (logit-transformed Platt’s scaling, beta calibration, logistic GAMs) are general-purpose calibration tools not specifically designed for this problem. The work reads more like a comprehensive empirical comparison of existing calibration methods under one scenario than a targeted solution to undersampling-induced miscalibration. This weakens the practical motivation, as it is unclear if undersampling plus post-hoc generic calibration is even the best approach compared to alternative strategies such as class weighting as what commonly random forest or boosted trees would do.

**[More model base validation]** Although the simulation study mentioned the behaviors of popular models like random forests, boosted trees, and Naive Bayes, the paper does not train or calibrate these real classifiers on undersampled data. Including such experiments on real data would better support the practical recommendations and demonstrate applicability in realistic ML workflows. Therefore, I am wondering if the authors would like to add more experiments about those model base to prove that it is the case for the real data.

**[More Real world dataset]** Although the wildfire prediction example is a promising illustration, I recommend that the authors also consider other types of highly imbalanced real-world datasets, such as rare disease prediction in medical data. Expanding the experimental validation to include diverse domains would make the paper more comprehensive, rather than relying solely on a single application.

**[Error bars for all experiments results]** Given the potential variance introduced by different undersampling strategies and retraining procedures, I think including error bars or confidence intervals across replicates is necessary. This would help demonstrate that the reported results are robust and that observed differences between calibration methods are statistically meaningful, rather than artifacts of random variation.

---

> ### Author Response · Authors · 2025-07-12
> **Response to Reviewer qpox (Weaknesses)**
>
> Thank you for your kind words and helpful criticism of our work. We have responded to individual comments on weaknesses here and on requested changes in the next response.
>
> Weaknesses:
>
> [Limited motivation]: You are correct that most of the calibration methods considered in our study are general purpose tools not specifically designed for calibration after undersampling. However, we are not aware of any calibration tools that have been specifically designed for this purpose. While the logit-transformed Platt’s scaling is a calibration method that has been used in the literature already for other purposes, we believe our work motivates its use for calibration after undersampling.
>
> We interpret the last sentence of your comment to mean the situation where models like random forests are trained on the entire dataset and then class weighting is used to undo the bias towards the majority class that is believed to exist in random forests (e.g., Chen et al., 2004). At this time, we do not wish to take a stance on whether this approach is better or worse than using undersampling. However, we note that Chen et al. (2004) studied both approaches for random forests and did not find that one performed better than the other. Undersampling has the advantage of using a much smaller subsample of the data, speeding up training time. It may be worthwhile for future work to do a more comprehensive comparison of undersampling plus post-hoc calibration and other methods. We intend to add some commentary on this in our Discussion in a revision.
>
> Chen, C., Liaw, A., & Breiman, L. (2004). Using random forest to learn imbalanced data. University of California, Berkeley, Department of Statistics Report.
>
> [More model base validation]: The miscalibration patterns of these models have been shown in previous work already (e.g., Niculescu-Mizil and Caruana, 2005). As we are already at the limit for a regular submission to TMLR, we do not wish to add several experiments. However, you are not the only reviewer to comment on including an experiment with a real model (aside from the GAM). We think there might be value in adding an experiment with a random forest, where we have indicated that traditional Platt’s scaling could be a good calibration technique to use. We intend to add this experiment in an appendix, using the wildfire occurrence data, and then comment on it in the main body of the paper.
>
> Niculescu-Mizil, A., & Caruana, R. (2005, August). Predicting good probabilities with supervised learning. In Proceedings of the 22nd International Conference on Machine Learning (pp. 625-632).
>
> [More real-world datasets/examples]: We agree with you that there are a number of real-world applications where this could be important. Topics such as natural hazards, rare diseases as you mentioned, fraud, suicidal ideation, and terrorism are all amenable to highly imbalanced datasets. Our goal with the wildfire prediction example was to motivate our work by showing that this calibration issue does impact at least one important area of study. However, the main contribution of our work lies in the simulation study we designed, so we do not believe that our paper is the best place for a more comprehensive exploration of the impacts of calibrating an undersampled dataset using Platt’s scaling in various fields, especially considering we are not experts in all of these fields. (We do have expertise in wildfire prediction.) Our hope is that our study will inspire people working in these other areas to consider the findings of our work in their modelling.
>
> [Error bars for all experiments results]: One of the other reviewers commented on finding the plots difficult to read. We believe that error bars will make this even more challenging, so we would like to avoid adding error bars. However, we agree with your point about showing that our findings are robust and not artifacts of random variation. Instead of error bars on the plots, we propose adding standard deviations for the RMSE and MAEs shown in the tables in the appendices. In addition, we can make our code available to readers so that they can download it and run it with any seed they want (or without seeds) so that they can see that our findings are not an artifact of random variation.

---

> > ### Author Response · Authors · 2025-07-12
> > **Response to Reviewer qpox (Requested Changes)**
> >
> > Requested Changes:
> >
> > [Scope Clarification] In the first paragraph of the current version of our manuscript, we attempted to justify why undersampling is used (i.e., to account for the class imbalance and large size of the datasets). Are there ways that you think we could emphasize this more? As mentioned above, we also plan to include some commentary on alternative strategies in our Discussion.
> >
> > [Add more model base] As we mentioned above, we are concerned about the length of the paper, but we think there could be value in adding the random forest model (with reliability plots showing its calibration on an undersampled testing dataset, as well as on a full testing dataset after calibration using various forms of Platt’s scaling) fit to the wildfire data. This way we don’t add to the paper excessively, but we handle a case where traditional Platt’s scaling might be useful.
> >
> > [More Real World Dataset Evaluation]: Please see our comments in the previous response for our thoughts on this issue.
> >
> > [Add Error bars]: Please see our comments in the previous response for our plan to address these concerns.

---

> > > ### Comment · Reviewer_qpox · 2025-07-15
> > > **Reply to Author**
> > >
> > > Thanks for all your replies about the review I have provided. I hope the modifications you have proposed will be reflected in the final version. Regarding the point about limited motivation, what I really mean is this: While I understand the focus on Platt’s scaling in the title, I notice that the paper actually offers a broader empirical comparison of multiple calibration methods (including beta calibration, isotonic regression, and logistic GAMs) in the context of undersampling. The simulations and conclusions discuss the relative strengths of all these methods. Therefore, I wonder if the writing focus might be improved by narrowing it more to emphasize Platt’s scaling itself towards this particular problem or make it broader to do as a comparison among different calibration tools for this task.

---

> > > > ### Author Response · Authors · 2025-07-16
> > > > **Response to Reviewer qpox**
> > > >
> > > > Thank you for your prompt reply to our comments, and we will begin implementing the modifications that we have proposed. Thank you as well for clarifying your concern about limited motivation. We will keep this in mind when working on the revised version of the manuscript.

---

### Review · Reviewer_qAL4 · 2025-07-04

**Summary Of Contributions:**

This paper studies when Platt’s scaling works, and when it fails, as a method of calibrating undersampled base models. Specifically, 4 paradigms are studied: when the base model is perfect, when it exhibits a bias towards 0.5 probability predictions, when it exhibits a bias towards extreme values, and when the base model makes very noisy predictions. The paper then provides theory to show that Platt’s fails in the case of a perfect model–but can be corrected with a simple transform–but that Platt’s is able to calibrate base models with a bias towards predicting probabilities of 0.5. Empirical experiments are then performed to validate the theory, and investigate the extreme-value bias and high noise cases. In summary, the paper provides nice insights into when Platt’s is appropriate for calibrating undersampled models.

**Audience:**

Yes

**Broader Impact Concerns:**

Given the increasing use of statistical learning methods not only for societally beneficial applications (like wildfire detection) but also for damaging ones (like weapons or surveillance systems), it would be nice to discuss some of the negative applications and steps that members of the STEM community might take to mitigate them. E.g. is calibration of an under-sampled base model a method that could be used for something like the Lavender targeting system? If yes, are there related research areas that we should be avoiding, or political or regulatory choices for which we should be advocating?

**Claims And Evidence:**

Yes

**Requested Changes:**

I have organized the requested changes into ‘Main Changes’, ‘Moderate Edits and Questions’, and ‘Minor Edits’. Respectively, these are changes requiring new figures/tables or edits to existing ones, smaller edits and clarifications, and typos/grammar changes. At the end of each bullet point in the first two sections I list whether it is critical (C) for my recommendation of acceptance, a suggestion to strengthen the paper (S), or just a clarification question (Q). All minor edits are (S).

## Main Changes

- Given the study of smaller-size synthetic datasets, it seems that some degree of variance is present in the RMSE and MAE values in the appendix. For this reason, it would be really nice to resample each dataset a handful of times, to come up with some measure of variation to include in the RMSE and MAE tables (e.g., include +/- 1 standard deviation). This would be particularly important to put to rest questions like the one discussed on page 9: “One unexpected result was that Platt’s scaling with the logit transformation (with or without the GAM) performed best in terms of RMSE when the mean outcome success rate was 0.0022 and 100 000 observations were in the calibration dataset. This may simply be due to randomness in the simulation process, however, as traditional Platt’s scaling outperformed Platt’s scaling with the logit transformation in all other cases, including when the mean outcome success rate was the same but more observations were used for calibration.” It could be nice to re-generate the plots using all the same methods but with a re-sampled dataset”. (S)
- I appreciate the interpretability, and clean-ness of the results on the synthetic dataset that one sees when using the ground truth (undersampled) base model ($\gamma$). However, it would be really nice to add experiments checking how cleanly the theory applies when (1) the base model is learned, and (2) the synthetic dataset is slightly more sophisticated. For example, it would be great to see simulations for a synthetic dataset with a more nonlinear relationship between covariates and probability (compared with the nonlinearity appearing only in the logit, with the current dataset), and to see undersampled base models that are trained, for example an MLP. Ideally, one would then also compare with models that are known to exhibit some of the systematic biases studied (e.g. random forests and/or Naive Bayes). (C)

## Moderate Edits and Questions

- Intro: given modern interest in probabilistic/Bayesian deep learning models (see work by, e.g., Yarin Gal, Zoubin Ghahramani), it could be worth at least touching on the relevance of the paper’s results to calibrating probabilities in deep learning. (S)
- Section 2: in the intro, the contents of section 2 are described: “we outline the calibration approaches considered in this study”, which, on first reading, suggested to me that it would be primarily a background section. However, in addition to a nice summary of calibration methods, it appears that some of the main theoretical results are also provided in this section (Theorems 1 and 2). It could be nice to alert the reader to this in advance, for greater clarity. E.g. ‘in section 2 we outline the calibration approaches considered in this study and provide our primary theoretical results (Theorems 1 and 2)’. (S)
- Section 2.2 lines 2-4: it would be good to provide a bit more of a discussion of the change to the output $y$ values, for regularization, suggested in the original paper, and to explain if/how the current results would be affected by utilizing this output change. This could be added in the appendix, and referenced in the main paper, if it does not fit nicely in main paper. (C)
- Paragraph 2, page 6: it is mentioned that, in the plots, only bins with more than 50 observations are retained. It would be nice to mention what fraction of the total data was removed by this protocol and, ideally, it would be good to show the plots with all the data points retained in the supplementary section, perhaps with different colours to denote the points that were removed. Similarly, it would be nice to at least give an idea of the number of bins that are dropped due to fewer than 10 observations in the plots later in the paper with synthetic data. (C)
- Figure 2: presumably the logit transform’s slight overprediction relative to the intercept adjustment is just due to noise? (Q)
- Section 4.1.1: any rationale for the dataset parameter choices (success probabilities and sample sizes)? Would be nice to provide a little bit of justification. (C)
- Simulation plots: it could be nice to make all the plots in the body of the paper have the same probability of success, so that one can better compare differences due to the biases (or lack thereof) in the base model. (S)
-Section 4.2.3 lines 5-6: “Isotonic regression is the only calibration method that did not exhibit visible systematic biases with this base model.” Thoughts on why this is the case? These could be useful to add. (S)
- In main plots: to clarify, all lines are plotted up to the last point at which there are no more bins with more than 10 (50) observations for the synthetic data (wildfire data)? (Q)

## Minor Edits
- Intro, paragraph 4, line 6; grammar: “models learned” => ‘models were learned’
- Line 5 page 4: change “...is intuitive that Platt’s scaling is unable to calibrate a model that is perfect…” to ‘is intuitive that Platt’s scaling is unable to calibrate model that are perfect…’, to be consistent with the plural on line 6, in “such models”
- Lines 1 and 2 after equation 5: what do you mean by “Mathematically, we require only that $r \in \mathbb{R}$ and $m \in [0, 1]$, but it is worth noting that settings within these bounds may not lead to a reasonable representation of a model.”?
- Line 4 on page 5: “generalized additive model (GAM)” => ‘Generalized Additive Model (GAM)’. I.e., capitalize for acronym.
- Line 1 after equation 7: “As mentioned in Section 2.3” is written, but this is contained in section 2.3! I suggest rewriting this to avoid referencing the current section.
- Line 2, section 2.4: subscript $i$ is used and summed over without definition (presumably one will realize that it’s a sum over samples but it could be nice to spell it out)

**Strengths And Weaknesses:**

## Strengths

The theoretical insights of the paper are elegant, and seem to be useful given how Platt’s scaling has been used in past literature. The paper is, for the most part, clearly written and logically structured–it was a pleasure to read. The plots in the experimental section are nicely arranged, and the choice of a wildfire dataset seems relevant given the climate crisis. I checked some of the paper’s references and it appears that Platt’s method has been used multiple times within the last two decades to calibrate undersampled models. Thus, if the authors’ claim that Platt’s has not been theoretically studied in the context of undersampling is indeed true, then this paper seems like a valuable and timely contribution. However, I must add the disclaimer that model calibration is not my area of expertise and, therefore, I am not the best judge of related literature and how the current paper is situated within it.

## Weaknesses

In my view, the two main weaknesses of the paper can, mostly, be straightforwardly addressed by a few extra simulation experiments. First, it would be nice to see some measure of variability for the quantitative results in the appendix–to provide standard deviations for the RMSE and MAE tables (see point 1 of Requested Changes-Main Changes). Second, it would be ideal if the authors could add a figure or two showing how clean the results are when the undersampled base model is not the ground truth, but a trained model, and when the synthetic dataset is more nonlinear (see point 1 of Requested Changes-Main Changes).

Finally, there are a couple experimental choices that could be better justified or elaborated upon (e.g. values selected for generating the synthetic dataset; dropping low-sample bins in certain figures). These issues are outlined in the requested changes and, I believe, should be relatively easy to remedy.

---

> ### Author Response · Authors · 2025-07-12
> **Response to Reviewer qAL4 (Part 1)**
>
> Thank you for your detailed and very organized review. With regards to your minor edits, we intend to implement all the suggested changes. Please see our comments below addressing individual concerns. Our response was more than 5000 characters so we have split it into multiple replies.
>
> Comment: In my view, the two main weaknesses of the paper can, mostly, be straightforwardly addressed by a few extra simulation experiments. First, it would be nice to see some measure of variability for the quantitative results in the appendix–to provide standard deviations for the RMSE and MAE tables (see point 1 of Requested Changes-Main Changes).
> Given the study of smaller-size synthetic datasets, it seems that some degree of variance is present in the RMSE and MAE values in the appendix. For this reason, it would be really nice to resample each dataset a handful of times, to come up with some measure of variation to include in the RMSE and MAE tables (e.g., include +/- 1 standard deviation). It could be nice to re-generate the plots using all the same methods but with a re-sampled dataset. (S)
>
> Response: Another reviewer had similar concerns about the potential variability in our results. We plan to address these concerns by using your suggestion of regenerating datasets several times. We plan to report the RMSE and MAE for the datasets visualized in the paper (as we have done already), as well as the mean and standard deviation across runs. To avoid repeating very similar plots within the paper, we do not want to include the same plots across iterations of each dataset. However, we intend to make the code available to readers so that they could easily generate plots themselves.
>
> Comment: Second, it would be ideal if the authors could add a figure or two showing how clean the results are when the undersampled base model is not the ground truth, but a trained model, and when the synthetic dataset is more nonlinear (see point 1 of Requested Changes-Main Changes).
> I appreciate the interpretability, and clean-ness of the results on the synthetic dataset that one sees when using the ground truth (undersampled) base model (γ). However, it would be really nice to add experiments checking how cleanly the theory applies when (1) the base model is learned, and (2) the synthetic dataset is slightly more sophisticated. For example, it would be great to see simulations for a synthetic dataset with a more nonlinear relationship between covariates and probability (compared with the nonlinearity appearing only in the logit, with the current dataset), and to see undersampled base models that are trained, for example an MLP. Ideally, one would then also compare with models that are known to exhibit some of the systematic biases studied (e.g. random forests and/or Naive Bayes). (C)
>
> Response: Another one of the reviewers asked us to include more models for the wildfire prediction example. Keeping in mind the length of the paper, we have suggested that we could include a detailed analysis of using a random forest in this setting, including reliability plots for the model on an undersampled testing dataset and on a full dataset after calibration. Would that be satisfactory for illustrating results with a real model?
>
> Comment: Could you please provide more information about what you would like to see in terms of non-linearity in the synthetic dataset?
>
> Response: The data generating process we have created already has five two-way interactions and two four-way interactions, so it is already quite non-linear. Is there a specific form of non-linearity you are interested in?
>
> Comment: Intro: given modern interest in probabilistic/Bayesian deep learning models (see work by, e.g., Yarin Gal, Zoubin Ghahramani), it could be worth at least touching on the relevance of the paper’s results to calibrating probabilities in deep learning. (S)
>
> Response: We intend to add a comment about this in our Introduction, emphasizing that undersampling can be valuable when fitting very complex models.
>
> Comment: Section 2: in the intro, the contents of section 2 are described: “we outline the calibration approaches considered in this study”, which, on first reading, suggested to me that it would be primarily a background section. However, in addition to a nice summary of calibration methods, it appears that some of the main theoretical results are also provided in this section (Theorems 1 and 2). It could be nice to alert the reader to this in advance, for greater clarity. E.g. ‘in section 2 we outline the calibration approaches considered in this study and provide our primary theoretical results (Theorems 1 and 2)’. (S)
>
> Response: This is a very good point. We will make changes to the Introduction in line with your suggestion.

---

> > ### Author Response · Authors · 2025-07-12
> > **Response to Reviewer qAL4 (Part 2)**
> >
> > Comment: Section 2.2 lines 2-4: it would be good to provide a bit more of a discussion of the change to the output y values, for regularization, suggested in the original paper, and to explain if/how the current results would be affected by utilizing this output change. This could be added in the appendix, and referenced in the main paper, if it does not fit nicely in main paper. (C)
> >
> > Response: The regularization would have little effect on the results in the paper because it is a small sample adjustment. For the cases with y=1, the target is changed to [sum(y_i) + 1]/[sum(y_i) + 2], which is approximately 1 for sufficiently large samples. An analogous adjustment is made for the cases with y=0. We will add these details in an appendix.
> >
> > Comment: Paragraph 2, page 6: it is mentioned that, in the plots, only bins with more than 50 observations are retained. It would be nice to mention what fraction of the total data was removed by this protocol and, ideally, it would be good to show the plots with all the data points retained in the supplementary section, perhaps with different colours to denote the points that were removed. Similarly, it would be nice to at least give an idea of the number of bins that are dropped due to fewer than 10 observations in the plots later in the paper with synthetic data. (C)
> >
> > Response: It is an extremely small fraction of the total data that was removed. We will add the exact numbers in a revision. We will also add plots in the supplementary material showing all of the data.
> >
> > Comment: Figure 2: presumably the logit transform’s slight overprediction relative to the intercept adjustment is just due to noise? (Q)
> >
> > Response: Yes, we have no reason to believe this is a systematic difference.
> >
> > Comment: Section 4.1.1: any rationale for the dataset parameter choices (success probabilities and sample sizes)? Would be nice to provide a little bit of justification. (C)
> >
> > Response: We wanted to consider fairly large datasets, as this is when undersampling provides the most benefit. To test the effect of the size of the calibration dataset, we just changed the size by an order of magnitude. For the success probabilities, we wanted to consider various levels of class imbalance. In our view, the probabilities that we’ve chosen reflect degrees of imbalance that we would call “imbalanced”, “very imbalanced”, and “extremely imbalanced” (although these are of course very subjective labels).
> >
> > Comment: Simulation plots: it could be nice to make all the plots in the body of the paper have the same probability of success, so that one can better compare differences due to the biases (or lack thereof) in the base model. (S)
> >
> > Response: Our thinking in the original version of the manuscript was that it would be nice to provide the reader some results from each of the different settings. Did you find this provided any value as you were reading it? If not, we’re happy to make the change you’ve suggested.
> >
> > Comment: Section 4.2.3 lines 5-6: “Isotonic regression is the only calibration method that did not exhibit visible systematic biases with this base model.” Thoughts on why this is the case? These could be useful to add. (S)
> >
> > Response: Isotonic regression is the most flexible calibration method, with no assumptions about the shape of the relationship (outside of it being monotonically increasing). This is why it does not have a systematic bias here. We can add this explanation in a revision.
> >
> > Comment: In main plots: to clarify, all lines are plotted up to the last point at which there are no more bins with more than 10 (50) observations for the synthetic data (wildfire data)? (Q)
> >
> > Response: The procedure of not plotting some observations only applies to the reliability plots, so just for the plots for the wildfire data and the nearly perfect model with the synthetic data. The other plots do not have bins, so all observations are plotted in these cases.

---

> > > ### Author Response · Authors · 2025-07-12
> > > **Response to Reviewer qAL4 (Part 3 - Broader Impact Concerns)**
> > >
> > > Comment: Given the increasing use of statistical learning methods not only for societally beneficial applications (like wildfire detection) but also for damaging ones (like weapons or surveillance systems), it would be nice to discuss some of the negative applications and steps that members of the STEM community might take to mitigate them. E.g. is calibration of an under-sampled base model a method that could be used for something like the Lavender targeting system? If yes, are there related research areas that we should be avoiding, or political or regulatory choices for which we should be advocating?
> > >
> > > Response: This is a difficult question for us to comment on, as we are not very familiar with the Lavender targeting system. With that said, we want to make it clear that the calibration transformations we have imposed are monotonic (with the exception of the GAM-based methods, but they will still learn nearly monotonic functions if the base model is good), so calibration has very little to no impact on the discriminatory ability of the model. Metrics like area under the receiver operating characteristic curve would remain unchanged, as would accuracy as long as different classification thresholds are used for the pre- and post-calibration models. In particular, improving calibration may not correct problems of fairness, and certainly won't correct for situations where models are deployed in situations and manners that are harmful.
> > >
> > > Our paper is primarily focused on the bias that can occur from using Platt’s scaling to calibrate after undersampling. Ultimately, it is these predictions that are used to make decisions, so this bias can lead to suboptimal decisions. Depending on the application, a positive bias may be more problematic than a negative bias, and vice-versa. For example, a slight positive bias in wildfire occurrence predictions, although not ideal, would be better than a slight negative bias because it could lead to more risk averse decision making by fire management staff. However, in other contexts, such as the Lavendar targeting system, overestimating probabilities may be the more problematic outcome. Again, improving calibration still does not fix situations where models are deployed in a harmful manner.

---

> > > > ### Comment · Reviewer_qAL4 · 2025-07-15
> > > > **response to broader impact concerns**
> > > >
> > > > Thank you for the thoughtful and detailed response! With my original comment I was particularly interested in the authors’ thoughts on how ML researchers might begin to address this latter question of when the “dual-use” (usable for positive and negative purposes) technologies they develop are applied in a harmful manner. While this question is quite general and could likely be asked of any paper submitted to TMLR, I believe it warrants some consideration given the severe impacts that models can have when applied harmfully, or when model development and implementation does not respect sustainability considerations, workers rights, or intellectual property rights.

---

> > > > > ### Author Response · Authors · 2025-07-16
> > > > > **Response to Reviewer qAL4**
> > > > >
> > > > > Thank you for your prompt response to our comments. We're happy to see that we can address your concerns with what we have proposed. With regards to the broader impact concerns, we completely agree with you that it is important to think of the dual use of methods we develop and consider how to ensure (to the best of our ability) that the methods are not applied in a harmful manner.

---

> > > ### Comment · Reviewer_qAL4 · 2025-07-15
> > > **response to authors (part 2)**
> > >
> > > The format below is the same as in part 1.
> > >
> > > ## Response to Author Responses
> > >
> > > "Response: It is an extremely small fraction of the total data that was removed. We will add the exact numbers in a revision. We will also add plots in the supplementary material showing all of the data."
> > >
> > > **Perfect, this is great**
> > >
> > > "Response: Our thinking in the original version of the manuscript was that it would be nice to provide the reader some results from each of the different settings. Did you find this provided any value as you were reading it? If not, we’re happy to make the change you’ve suggested."
> > >
> > > **While I might have a slight preference for keeping the same probability of success in the main body, this is probably more of a personal preference than anything else, and I can also see why you decided to change them. I am happy with whatever the authors choose on this point.**
> > >
> > > ## Summary
> > > As in part 1, if the authors make the changes that they outline in their responses then my concerns outlined in this section will have been sufficiently addressed.

---

> > ### Comment · Reviewer_qAL4 · 2025-07-15
> > **response to authors (part 1)**
> >
> > I have included responses, below, to some of the author responses to my original comments. The format is as follow:
> >
> > "snippet of author response to my original question..."
> >
> > **my new response in bold**
> >
> > ## Responses to author responses
> >
> > "Response: Another reviewer had similar concerns about the potential variability in our results. We plan to address these concerns by..."
> >
> > **This is great, thank you.**
> >
> > "Response: Another one of the reviewers asked us to include more models... Would that be satisfactory for illustrating results with a real model?"
> >
> > **This would be enough to address my concern, thanks!**
> >
> > "Response: The data generating process we have created already has five two-way interactions and two four-way interactions, so it is already quite non-linear. Is there a specific form of non-linearity you are interested in?"
> >
> > **I apologize, I had forgotten that you included several multiple-way interactions in the dataset. Please disregard this comment.**
> >
> > "Response: We intend to add a comment about this in our Introduction, emphasizing that undersampling can be valuable when fitting very complex models."
> >
> > **Great, thank you!**
> >
> > Comment: Section 2: in the intro, the contents of section 2 are described: “we outline the calibration approaches considered in this study”, which...
> > Response: This is a very good point. We will make changes to the Introduction in line with your suggestion.
> >
> > **Perfect!**
> >
> > ## Summary
> > If the authors make the changes that they outline in their responses then my concerns outlined in this section will have been sufficiently addressed.

---

### Decision · Action_Editor_c1XC · 2025-08-03

**Recommendation:** Accept with minor revision

**Additional Comments:**

This paper requires the following revisions before acceptance:

- Repeat the experiments on random forest outputs, so that the readers can see results on both ground truth data and a predictive model
- Add error bars around all of the results (on both ground truth data and random forest prediction experiments)
- Add missing experimental details to the text (e.g. why certain data points were dropped from the plots)
- Address other presentation issues from the reviewers (e.g. line colours on plots, etc.)

**Audience:**

Yes

**Audience Explanation:**

Calibration is an issue of interest to the TMLR community, and Platt scaling is one of the most commonly used post-hoc calibration methods. Given the prevalence of imbalanced datasets in machine learning applications, the topic of this paper will be of interest to the community. As one reviewer noted, Platt's method has been used multiple times within the last two decades to calibrate undersampled models, making this analysis both timely and practically relevant.

This paper could have garnered even greater interest and significance if the authors had expanded their scope in two key areas. First, broadening the case study beyond Platt scaling variants to include a more comprehensive comparison of calibration methods would have provided stronger practical recommendations. Second, expanding the empirical evidence with datasets from other domains beyond wildfire prediction would have demonstrated broader applicability. Reviewers specifically requested both of these expansions, but the authors declined to address these opportunities, citing constraints on paper length and their domain expertise. Nevertheless, the paper in its current focused scope meets the audience interest threshold for TMLR.

**Claims And Evidence:**

No

**Claims Explanation:**

This paper focuses on Platt scaling and other post-hoc calibration methods on imbalanced datasets. The authors make two primary contributions: theoretical analysis demonstrating when Platt's scaling fails on undersampled datasets, and an empirical investigation of Platt's scaling and related post-hoc calibration methods using simulation data in this setting. The theoretical work is grounded and correct, while the simulation study provides characterization of post-hoc calibration methods across a variety of noise and imbalance scenarios on a wildfire prediction dataset.

There are several outstanding concerns about the current evidence provided by the authors. The simulation study calibrates ground-truth probability data rather than outputs from actual machine learning models. Additionally, there are technical issues with the empirical validation: the authors lack error bars around simulation results, do not adequately explain data points dropped from plots, and provide insufficient detail about experimental procedures. The authors have committed to addressing these limitations by including a calibration study on random forest predictions, repeating simulations to report variance measures, and providing clearer explanations of their data handling procedures. **My recommendation for acceptance is conditional on the authors implementing these promised changes in their revised manuscript.** Until these improvements are made, the evidence supporting the practical claims remains incomplete.

---

> ### Author Response · Authors · 2025-08-09
> **Response to Decision**
>
> Thank you for taking the time to assess our work and the overall positive decision.
>
> We had submitted a revision on July 29th, but it seems like we accidentally submitted the original file rather than the revised version. We have just now created a second revision with the file we intended to submit on July 29. In this file, we make the revisions that we said we intended to complete in our responses to the reviewers.
>
> However, we have not quite addressed all the revisions requested in the decision. We make individual comments on these below:
>
> > Repeat the experiments on random forest outputs, so that the readers can see results on both ground truth data and a predictive model
>
> We have not repeated all experiments using random forests. We have fit a random forest to the wildland fire data, not the synthetic data.
>
> > Add error bars around all of the results (on both ground truth data and random forest prediction experiments)
>
> We have not added error bars to our plots due to concerns with viewing/interpreting the plots if error bars were added. We have added confidence intervals to the tables to address concerns about variability of the results. We also do not have error bars for the random forest results, but we do have 95% prediction intervals.
>
> > Add missing experimental details to the text (e.g. why certain data points were dropped from the plots)
>
> We have added experimental details and included plots in the appendix with the removed data points.
>
> > Address other presentation issues from the reviewers (e.g. line colours on plots, etc.)
>
> We have addressed the concerns with presentation.
>
> We would just like to clarify if the changes that we have made in this revision are sufficient for acceptance or if we are being asked to add more content. Our impression from the reviewers' responses was that the revisions we have now made would address their concerns, but we are not sure if the editor would like to see some of the other suggestions implemented anyway. Thank you.

---

> > ### Comment · Action_Editor_c1XC · 2025-08-14
> >
> > Thank you for explaining your choices! I have looked through the revision and I am satisfied with the changes, and will therefore recommend acceptance.